# Off-Dynamics Reinforcement Learning via Domain Adaptation and Reward Augmented Imitation

**Yihong Guo[1], Yixuan Wang[1], Yuanyuan Shi[2], Pan Xu[3], Anqi Liu[1]**
[1]Johns Hopkins University
[2]University of California San Diego
[3]Duke University
{yguo80,ywang830,aliu.cs}@jhu.edu, yyshi@ucsd.edu, pan.xu@duke.edu

## Abstract

Training a policy in a source domain for deployment in the target domain under a dynamics shift can be challenging, often resulting in performance degradation. Previous work tackles this challenge by training on the source domain with modified rewards derived by matching distributions between the source and the target optimal trajectories. However, pure modified rewards only ensure the behavior of the learned policy in the source domain resembles trajectories produced by the target optimal policies, which does not guarantee optimal performance when the learned policy is actually deployed to the target domain. In this work, we propose to utilize imitation learning to transfer the policy learned from the reward modification to the target domain so that the new policy can generate the same trajectories in the target domain. Our approach, *Domain Adaptation and Reward Augmented Imitation Learning* (DARAIL), utilizes the reward modification for domain adaptation and follows the general framework of *generative adversarial imitation learning from observation* (GAIfO) by applying a reward augmented estimator for the policy optimization step. Theoretically, we present an error bound for our method under a mild assumption regarding the dynamics shift to justify the motivation of our method. Empirically, our method outperforms the pure modified reward method without imitation learning and also outperforms other baselines in benchmark off-dynamics environments.

## 1 Introduction

The objective of reinforcement learning (RL) is to learn an optimal policy that maximizes rewards through interaction and observation of environmental feedback. However, in domains such as medical treatment [1] and autonomous driving [2], we cannot interact with the environment freely as the errors are too costly or the amount of access to the environment is limited. Instead, we might have access to a simpler or similar source domain. This requires domain adaptation in reinforcement learning. In this paper, we study a specific problem of domain adaptation in reinforcement learning (RL), where only the dynamics (transition probability) are different in two domains. This is called *off-dynamics RL* [3–5]. Specifically, we focus on a problem setting in which we have limited access to rollout data from the target domain, but we do not have access to the target domain reward, following the previous off-dynamics work [3–5].

Previous work on off-dynamics RL, such as *Domain Adaptation with Rewards from Classifiers* (DARC) [3] and [6, 5], focuses on training the policy in the source domain with a modified reward function that compensates for the dynamics differences. The reward modification is derived so that the distribution of the learning policy's experience in the source domain matches that of the optimal trajectories in the target domain. As a result, their experience in the source domain will

38th Conference on Neural Information Processing Systems (NeurIPS 2024).

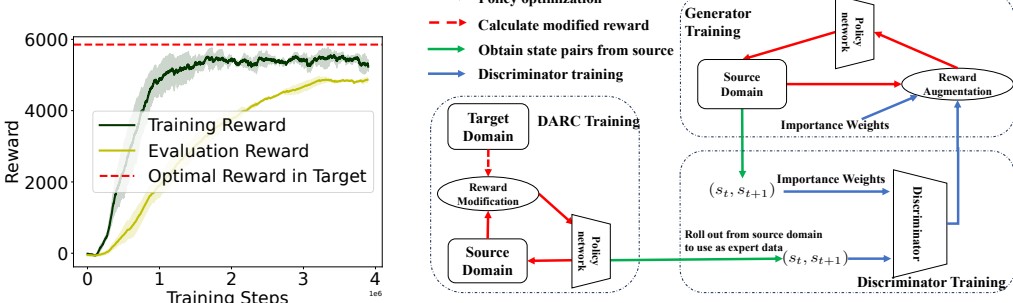

Figure 1: (a) Training reward in the source domain, i.e. $\mathbb{E}_{\pi_{\text{DARC}}, p_{\text{src}}}[\sum_t r(s_t, a_t)]$, evaluation reward in the target domain, i.e. $\mathbb{E}_{\pi_{\text{DARC}}, p_{\text{trg}}}[\sum_t r(s_t, a_t)]$ and optimal reward in target domain, for DARC in Ant. Evaluating the trained DARC policy in the target domain will cause performance degradation compared with its training reward, which should be close to the optimal reward in the target given DARC's objective function. Results of HalfCheetah, Walker2d, and Reacher are in Figure 9 in Appendix. (b) Learning framework of DARAIL. DARC Training: we first train the DARC in the source domain with a modified reward that is derived from the minimization of the reverse divergence between optimal policies on target and learned policies on the source. Details of DARC and the modified reward are in Section 3.1 and Appendix A.1. Discriminator training: the discriminator is trained to classify whether the data is from the expert demonstration (DARC trajectories) and provide a local reward function for policy learning. Generator training: the policy is updated with augmented reward estimation, which integrates the reward from the source domain and information from the discriminator. We first train DARC, collect DARC trajectories from the source domain, and then train the discriminator and the generator alternatively.

produce a trajectory distribution close to the target domain's optimal one. However, deploying the resulting policy in the target domain usually causes performance degradation compared to its training performance in the source domain. Figure 1 (a) shows the experiment result of DARC under a broken source environment setting, where the broken source environment means the value of 0-index in the action of the source domain is frozen to 0, and the target environment remains intact. Consequently, existing reward modification methods will only obtain a sub-optimal policy in the target domain. Details of DARC and its suboptimality in the target domain will be introduced in Section 3.1. More details about why DARC fails in more general dynamics shift cases are in Appendix C.6.

In this paper, we present an off-dynamics reinforcement learning algorithm described in Figure 1 (b). Our method, Domain Adaptation and Reward Augmented Imitation Learning (DARAIL) consists of two components. Following previous work like DARC [3] on off-dynamics RL, we first obtain the source domain trajectories that resemble the target domain's optimal ones. We then transfer the policy's behavior from the source to the target domain through imitation learning from observation [7], which can mimic the policy's behavior from the state space.

In particular, we consider the dynamics shift in the framework of generative adversarial imitation from observation (GAIfo) [8], and propose a novel and practical reward estimator called the *reward augmented estimator* ($R_{AE}$) for the policy optimization step in imitation learning.

**Our contributions** can be summarized as follows:

- We propose the Domain Adaptation and Reward Augmented Imitation Learning (DARAIL) algorithm by transferring the learned policy of reward modification approaches from the source domain to the target domain via mimicking state-space trajectories in the source domain. We propose *reward augmented estimator* ($R_{AE}$) to leverage the reward from the source domain to stabilize the learning.
- We recognize limitations in the existing DARC algorithm and off-dynamics reinforcement learning algorithms with similar reward modification, which is directly deploying the learned policy to the target domain results in significant performance degradation. Our proposed algorithm mitigates this issue with an imitation learning component that transfers DARC policy to the target.
- We introduce an error bound for DARAIL that relaxes the assumption made in previous works that the optimal policy will receive a similar reward in both domains. Specifically, with our imitation

learning from the observation component, we can show the convergence of DARAIL with a mild assumption on the magnitude of the dynamics shift.

- We conducted experiments on four Mujoco environments, namely, *HalfCheetah*, *Ant*, *Walker2d*, and *Reacher* on modified gravity/density configurations and broken action environments. A comparative analysis between DARAIL and baseline methods is performed, demonstrating the effectiveness of our approach. Our method exhibits superior performance compared to the pure modified reward method without imitation learning and outperforms other baselines in these environments. Code is available at https://github.com/guoyihonggyh/Off-Dynamics-Reinforcement-Learning-via-Domain-Adaptation-and-Reward-Augmented-Imitation.

## 2   Backgrounds

**Off-dynamics reinforcement learning** We consider two Markov Decision Processes (MDPs): one is the source domain $\mathcal{M}_{\text{src}}$, defined by $(\mathcal{S}, \mathcal{A}, \mathcal{R}, p_{\text{src}}, \gamma)$, and the other one is the target domain $\mathcal{M}_{\text{trg}}$, defined by $(\mathcal{S}, \mathcal{A}, \mathcal{R}, p_{\text{trg}}, \gamma)$. The difference between them is the dynamics $p$, also known as transition probability, i.e., $p_{\text{src}} \neq p_{\text{trg}}$ or $p_{\text{src}}(s_{t+1}|s_t, a_t) \neq p_{\text{trg}}(s_{t+1}|s_t, a_t)$. In our paper, we experiment with two types of dynamics shift: 1) broken environment [3], in which the 0-th index value is set to be 0 in action, and 2) modifying the gravity/density setting of the target environment [9]. The source and the target domain share the same reward function, i.e., $r_{\text{src}}(s_t, s_{t+1}) = r_{\text{trg}}(s_t, s_{t+1})$. All other settings, including state space $\mathcal{S}$, action space $\mathcal{A}$, and the discounting factor $\gamma$, are the same. We will use $\gamma = 1$ in the derivation and analysis in our paper.

We aim to learn a policy $\zeta(a|s)$ using interaction from the source domain together with a small amount of data from the target domain $(s_t, a_t, s_{t+1})_{\text{trg}}$ to maximize the expected discounted sum of reward $\mathbb{E}_{\zeta, p_{\text{trg}}}[\sum_t \gamma^t r(s_t, a_t)]$ in the target domain. Note that we assume we only have limited access to the target domain transition, namely $(s_t, a_t, s_{t+1})_{\text{trg}}$, in the whole process and we do not utilize the target domain reward.

**Imitation learning (from Observation)** Imitation Learning (IL) trains a policy to mimic an expert policy $\pi_E$ with expert demonstration $\{(s_0, a_0), (s_1, a_1), ...\}$ or $\{(s_0, s_1), (s_1, s_2), ...\}$. Generative adversarial imitation learning (GAIL) [7] uses an objective similar to Generative adversarial networks (GANs) that minimizes the distribution generated by the policy and the expert demonstration. It alternatively trains a discriminator $D_\omega$ and a policy $\pi_\theta$ to solve the min-max problem:

$$\min_{\pi_\theta} \max_{D_\omega} \mathbb{E}_{(s,s')\sim\pi_E}\big[\log D_\omega(s, s')\big] + \mathbb{E}_{(s,s')\sim\pi_\theta}\big[\log(1 - D_\omega(s, s'))\big] - \lambda\mathcal{H}(\pi_\theta), \quad (2.1)$$

where $s'$ is the next state and $\mathcal{H}(\pi_\theta)$ is the entropy of the policy $\pi_\theta$. Note that in our problem, we mimic the state-only expert demonstrations $\{(s_0, s_1), (s_1, s_2), ...\}$ instead of the expert's actions. This setting is also called imitation learning from observation [8]. We will further discuss why we use state observation instead of action in section 3.2. $D_\omega$ is the classifier that discriminates whether the state pair is from the expert $\pi_E$ or generated by the policy $\pi_\theta$. Then, the policy is trained with the RL algorithm using reward estimation $-\log D_\omega(s, s')$ as the reward. The optimization of the Eq. (2.1) involves alternatively training the policy and the discriminator.

## 3   Off-dynamics RL via Domain Adaptation and Reward Augmented Imitation Learning

In this section, we present our algorithm, DARAIL, under the off-dynamics RL problem setting. First, we introduce DARC [3] in Section 3.1, which provides the distribution of target optimal trajectories in the source domain to mimic. Then, in Section 3.2, we introduce the imitation learning component through which we utilize the trajectories provided by DARC and transfer the DARC policy to the target domain. We aim to learn a policy that generates the same distribution of trajectories in the target domain as the DARC trajectories in the source domain.

### 3.1   Off-dynamics RL via Modified Reward

DARC is proposed to solve the off-dynamics RL through a modified reward that compensates for the dynamics shift [3]. Here, we first introduce DARC and its drawbacks. DARC seeks to match the policy's experiences in the source domain and optimal trajectories in the target domain. We

define $\tau = \{(s_1, a_1), (s_2, a_2), ..., (s_t, a_t), ...\}$ as a trajectory. We use $\tau_{\pi_\theta}^{\text{src}}$ to represent the trajectories generated by $\pi_\theta$ in the source domain. The policy's distribution over trajectories in the source domain is defined as:

$$q(\tau_{\pi_\theta}^{\text{src}}) = p_1(s_1) \prod_t p_{\text{src}}(s_{t+1}|s_t, a_t)\pi_\theta(a_t|s_t). \tag{3.1}$$

Let $\pi^* = \text{argmax}_\pi \mathbb{E}_{\pi, p_{\text{trg}}} \left[ \sum_t r(s_t, a_t) \right]$ be the policy maximizing the cumulative reward in the target domain. We use $\tau_{\pi^*}^{\text{trg}}$ to represent the trajectories generated by $\pi^*$ in the target domain. Given the assumption that the optimal policy $\pi^*$ in the target domain is proportional to the exponential reward, i.e., $\pi^*(a_t|s_t) \propto \exp(\sum_t r(s_t, a_t))$, the desired distribution over trajectories in the target domain is defined as:

$$p(\tau_{\pi^*}^{\text{trg}}) \propto p_1(s_1) \prod_t p_{\text{trg}}(s_{t+1}|s_t, a_t) \times \exp\left(\sum_t r(s_t, a_t)\right). \tag{3.2}$$

DARC policy can be obtained by minimizing the reverse KL divergence of $p(\tau_{\pi^*}^{\text{trg}})$ and $q(\tau_{\pi_\theta}^{\text{src}})$:

$$\min_{\pi_\theta} \mathcal{D}_{\text{KL}}(q||p) = -\min \mathbb{E}_{p_{\text{src}}} \sum_t r(s_t, a_t) + \Delta r(s_t, a_t, s_{t+1}) + \mathcal{H}_{\pi_\theta}[a_t|s_t] + c, \tag{3.3}$$

where $\Delta r(s_t, a_t, s_{t+1}) := \log p_{\text{trg}}(s_{t+1}|s_t, a_t) - \log p_{\text{src}}(s_{t+1}|s_t, a_t)$ and $c$ is a partition function of $p(\tau_{\pi^*}^{\text{trg}})$, which is independent of the dynamics and policy. The $\Delta r(s_t, a_t, s_{t+1})$ can be calculated through the following procedure: i), train two classifiers $p(\text{trg}|s_t, a_t)$ and $p(\text{trg}|s_t, a_t, s_{t+1})$ with cross-entropy loss $\mathcal{L}_{CE}$; ii), Use Bayes' rules to obtain the $\log\left(\frac{p_{\text{trg}}(s_{t+1}|s_t, a_t)}{p_{\text{src}}(s_{t+1}|s_t, a_t)}\right)$. Details are in Appendix C.1. Eq. (3.3) shows that $\pi_{\text{DARC}}$ can be obtained via maximum entropy algorithm with a modified reward $r_{\text{modified}} = r(s_t, a_t) + \Delta r(s_t, a_t, s_{t+1})$ at every step.

However, DARC matches the distribution of $\tau_{\pi^*}^{\text{trg}}$ and $\tau_{\pi_{\text{DARC}}}^{\text{src}}$. As the dynamics shift exists, $\pi_{\text{DARC}}$ will not recover the optimal policy $\pi^*$, and deploying the DARC in the target domain will usually suffer from performance degradation due to the dynamics shift, as shown in Figure 1(a) and Figure 9 in Appendix. However, in the source domain $\tau_{\pi_{\text{DARC}}}^{\text{src}}$ resembles those optimal trajectories in the target domain. Given the property of $\tau_{\pi_{\text{DARC}}}^{\text{src}}$, we propose to use imitation learning from observation with $\tau_{\pi_{\text{DARC}}}^{\text{src}}$ as expert demonstrations to transfer DARC to the target domain. The new policy in the target domain should behave similarly (generate similar trajectories) as DARC in the source domain.

## 3.2 Imitation Learning from Observation with Reward Augmentation

In this section, we present the *Domain Adaptation and Reward Augmented Imitation Learning* (DARAIL) method, which mitigates the problem of DARC via imitation learning from observation. As described in Section 3.1, $\tau_{\pi_{\text{DARC}}}^{\text{src}}$ resembles the target optimal trajectories, and we want to transfer DARC's behavior to the target domain. A natural way to tackle it is utilizing imitation learning to mimic the expert demonstration $\tau_{\pi_{\text{DARC}}}^{\text{src}}$. Following [7, 8], the objective can be formulated as:

$$\min_\zeta \max_{D_\omega} \left\{ \mathbb{E}_{p_{\text{trg}}, \zeta} \left[ \sum_t \log D_\omega(s_t, s_{t+1}) \right] + \mathbb{E}_{(s_t, s_{t+1}) \sim \tau_{\pi_{\text{DARC}}}^{\text{src}}} \left[ \sum_t \log(1 - D_\omega(s_t, s_{t+1})) \right] \right\}. \tag{3.4}$$

where $D_\omega$ is the discriminator in the generative adversarial imitation learning and $\zeta$ is the policy to be learned in the target domain. In the objective function Eq. (3.4), the $(s_t, s_{t+1})$ pairs are from the target domain, while we do not have much access to the target domain. Alternatively, we can use the $(s_t, s_{t+1})$ pairs from the source domain and re-weight the transition with the importance sampling method to account for the dynamics shift. The objective with data rolled out from the source domain, and the importance sampling is as follows:

$$\min_\zeta \max_{D_\omega} \left\{ \mathbb{E}_{p_{\text{src}}, \zeta} \left[ \sum_t \rho(s_t, s_{t+1}) \log D_\omega(s_t, s_{t+1}) \right] + \mathbb{E}_{(s_t, s_{t+1}) \sim \tau_{\pi_{\text{DARC}}}^{\text{src}}} \left[ \sum_t \log(1 - D_\omega(s_t, s_{t+1})) \right] \right\}, \tag{3.5}$$

where $\rho(s_t, s_{t+1}) = \frac{p_{\text{trg}}(s_{t+1}|s_t, a_t)}{p_{\text{src}}(s_{t+1}|s_t, a_t)}$ is the importance weight. Note that we do the generative adversarial imitation learning from only state observations (*GAILfo*) with $(s_t, s_{t+1})$ [9–11] instead of $(s_t, a_t)$. This is because we aim to learn a policy $\zeta$ to produce the same trajectory distributions in the target as the ones $\pi_{\text{DARC}}$ produces in the source domain, despite the dynamics shift, rather than mimicking the policy. Mimicking the $(s_t, a_t)$ pairs will recover the same policy as DARC, and deploying it to the target domain will not recover the expert trajectories due to the dynamics shift.

This objective Eq. (3.5) can be interpreted as training the discriminator $D_\omega$ to discriminate whether the $(s_t, s_{t+1})$ generated by $\zeta$ in the target domain matches the distribution of DARC trajectories

in the source domain using data rolled out from the source domain with $\zeta$ and importance weight. Then, after the discriminator is fitted, the policy can be trained with the reward estimator $R_{AE}$ with model-free RL. The objective is:

$$\max_\zeta \mathbb{E}_{p_{\text{src}},\zeta} \left[ \sum_t R_{AE}(s_t, s_{t+1}) \right], \qquad (3.6)$$

where $R_{AE}$ is defined as follows:

$$R_{AE}(s_t, s_{t+1}) = -\log D_\omega(s_t, s_{t+1}) + \rho(s_t, s_{t+1})(r_{\text{src}}(s_t, s_{t+1}) + \log D_\omega(s_t, s_{t+1})). \qquad (3.7)$$

Here the $r_{\text{src}}(s_t, s_{t+1})$ is the reward obtained from the source domain, which is the same as the reward from the source domain, i.e. $r_{\text{trg}}(s_t, s_{t+1})$. In imitation learning, the $-\log D_\omega(s_t, s_{t+1})$ can be viewed as a local reward function for the policy optimization step and the objective is $\max_\zeta \mathbb{E}_{p_{\text{src}},\zeta}[\sum_t -\log D_\omega(s_t, s_{t+1})]$. So Eq.(3.5) can be viewed as learning a reward function for the training of $\zeta$. However, as the dynamics shift exists, the estimation of the $-\log D_\omega(s_t, s_{t+1})$ could be biased, which is similar to the case in off-policy evaluation (OPE) [12–16] when training a reward estimation on biased data. As we have access to the source domain and can obtain the reward from the rollout, we are motivated to use both the reward estimation $-\log D_\omega(s_t, s_{t+1})$ and the ground truth reward in the source domain $r_{\text{src}}(s_t, s_{t+1})$ so that we could have a better reward estimation than $-\log D_\omega(s_t, s_{t+1})$ under dynamics shift. The $R_{AE}$ here can be viewed as using $-\log D_\omega(s_t, s_{t+1})$ as a base estimator of the reward and use $r_{\text{src}}(s_t, s_{t+1})$ and importance weight $\rho(s_t, s_{t+1})$ to correct it. This correction idea is similar to the doubly robust estimator (DR) [12] in OPE. The DR estimator combines the reward estimation $\hat{r}$ and the importance-weighted difference between true reward $r$ and $\hat{r}$. Specifically, the DR method takes the reward estimation $\hat{r}$ as a base estimator and applies the importance weighting to the difference between true reward $r$ and $\hat{r}$, which is $\rho(r - \hat{r})$ term, to correct the bias of the $\hat{r}$, where $\rho$ is the importance weight.

---

**Algorithm 1** Domain Adaptation and Reward Augmented Imitation Learning (DARAIL)

---

1: Initialize: source and target environments $\mathcal{M}_{\text{src}}$ and $\mathcal{M}_{\text{trg}}$; replay buffers for source and target transitions, $(\mathcal{D}_{\text{src}}^{\pi_{\text{DARC}}}, \mathcal{D}_{\text{trg}}^\zeta, \mathcal{D}_{\text{src}}^\zeta)$; initial parameters for the two classifiers $\theta = (\theta_{\text{SA}}, \theta_{\text{SAS}})$; initial policy $(\pi_{\text{DARC}}, \zeta)$; initial discriminator $D_\omega$, ratio r of experience from source vs. target, ratio k of update frequency of generator vs. discriminator.

2: $\pi_{\text{DARC}} \leftarrow$ Call DARC [3]          $\triangleright$ training expert policy

   *Reward Augmented Imitation Learning*

3: $\mathcal{D}_{\text{src}}^{\pi_{\text{DARC}}} \leftarrow \mathcal{D}_{\text{src}}^{\pi_{\text{DARC}}} \bigcup \text{ROLLOUT}(\pi_{\text{DARC}}, \mathcal{M}_{\text{src}})$

4: **for** $t = 0, ...T$ **do**

5:    $\mathcal{D}_{\text{src}}^\zeta \leftarrow \mathcal{D}_{\text{src}}^\zeta \bigcup \text{ROLLOUT}(\zeta, \mathcal{M}_{\text{src}})$

6:    **if** $t \bmod r = 0$ **then**

7:       $\mathcal{D}_{\text{trg}}^\zeta \leftarrow \mathcal{D}_{\text{trg}}^\zeta \bigcup \text{ROLLOUT}(\zeta, \mathcal{M}_{\text{trg}})$

8:    **end if**

9:    **if** $t \bmod k = 0$ **then**

10:       $D_\omega \leftarrow \text{IL}(\mathcal{D}_{\text{src}}^{\pi_{\text{DARC}}}, \mathcal{D}_{\text{src}}^\zeta, \mathcal{L})$, where $\mathcal{L}$ is from Eq. (3.5)    $\triangleright$ update discriminator

11:    **end if**

12:    $\theta \leftarrow \operatorname{argmin} \mathcal{L}_{\text{CE}}(\mathcal{D}_{\text{src}}^\zeta, \mathcal{D}_{\text{trg}}^\zeta)$      $\triangleright$ update classifiers by cross-entropy loss

13:    Calculate $R_{AE}$ from Eq.(3.7)           $\triangleright$ reward augmented estimator

14:    $\zeta \leftarrow \text{SAC}(\zeta, \mathcal{D}_{\text{src}}^\zeta, R_{AE})$            $\triangleright$ update generator

15: **end for**

16: **Output:** $\zeta$

---

**Our Algorithm** The DARAIL is shown in Algorithm 1, which consists of two steps: the first step, Line 2 in Algorithm 1, is the training of $\pi_{\text{DARC}}$, and the second step is imitation learning with the reward estimator in Eq. (3.7). In Lines 6-8, we roll out the target domain transition $(s_t, a_t, s_{t+1})$ to calculate the importance weight. Here, we will not collect the target domain reward. In Lines 9-11, we update the discriminator based on Eq. (3.5). In Line 12, we train the two classifiers $p(\text{trg}|s_t, a_t)$ and $p(\text{trg}|s_t, a_t, s_{t+1})$ with cross-entropy loss $\mathcal{L}_{CE}$ and Bayes' rules similar to $\Delta r(s_t, a_t, s_{t+1})$ in DARC as mentioned in Section 3.1. The details are in Appendix C.1. Lastly, we calculate the $R_{AE}$ in Line 13 and update the generator (Soft Actor-Critic (SAC) [17]) with $R_{AE}$ in Line 14.

Note that in Lines 6-7, we roll out from the target domain, but the amount of it is significantly smaller than the source rollouts. In our experiments, we roll out from the target domain every 100 steps of

source domain rollouts, which is 1% of the source domain rollouts. Further, even though DARAIL requires more target domain rollouts than DARC as it is required to train DARC first and then perform the imitation learning step, the advantage of DARAIL does not solely come from the more target samples. Because, in DARC, increasing the training step or target domain rollouts will not further improve its performance due to its inherent suboptimality, which is shown in table 11 and 12 in Appendix with the same amount of target domain rollouts.

## 4 Theoretical Analysis of DARAIL

Let $\pi^* = \arg\max_\pi \mathbb{E}_{\pi, p_{\text{trg}}} \left[ \sum_t r(s_t, a_t) \right]$ be the optimal policy maximizing the cumulative reward in the target domain and $\hat{\zeta}$ be the policy learned from DARAIL. Now, we provide an error bound for DARAIL. Details of the proof are deferred to Appendix B.

**Theorem 4.1.** *Let $m$ be the number of the expert demonstration and $\hat{\mathcal{R}}_\pi^{(m)} = \mathbb{E}_\sigma \left[ \sup_{D \in \mathcal{D}} \frac{1}{m} \sum_{i=1}^m \sigma_i D(s_t, s_{t+1}) \right]$ be the empirical Rademacher complexity. Let $B$ be the error bound of DARC in the source domain, i.e. $\mathbb{E}_{p_{src}, \pi^*_{DARC}} \left[ \sum_t r(s_t, a_t) + \mathcal{H}[a_t|s_t] \right] - \mathbb{E}_{p_{src}, \pi_{DARC}} \left[ \sum_t r(s_t, a_t) \right] \leq B$ and $W$ be the upper bound of the importance weight, i.e. $\rho(s_t, s_{t+1}) \leq W, \forall (s_t, s_{t+1})$. Let discriminator class $\mathcal{D}$ be a $\Delta$-bounded function, i.e. $|D_\omega(s_t, s_{t+1})| \leq \Delta$ given any $(s_t, s_{t+1})$. $\|r\|_\mathcal{D}$ measures the richness of the discriminator to represent the ground truth reward as defined in Appendix B.2. $d_\mathcal{D}$ is a defined neural network distance between the $(s_t, s_{t+1})$ distributions generated by the $\pi_{DARC}$ and $\pi_{\hat{\zeta}}$ defined in Appendix B.1. Given the empirical training error of the imitation learning, i.e. $d_\mathcal{D}(\hat{\tau}^{src}_{\pi_{DARC}}, \hat{\tau}^{trg}_{\hat{\zeta}}) - \inf_\zeta d_\mathcal{D}(\hat{\tau}^{src}_{\pi_{DARC}}, \hat{\tau}^{trg}_\zeta) \leq \hat{\epsilon}, \forall \delta \in (0, 1)$, with probability at least $1 - \delta$, we have*

$$\mathbb{E}_{p_{trg}, \pi^*} \left[ \sum_t r(s_t, a_t) \right] - \mathbb{E}_{p_{trg}, \hat{\zeta}} \left[ \sum_t r(s_t, a_t) \right]$$

$$\leq \underbrace{\mathbb{E}_{p_{src}, \pi^*_{DARC}} \left[ \sum_t r(s_t, a_t) + \mathcal{H}[a_t|s_t] \right] - \mathbb{E}_{p_{src}, \pi_{DARC}} \left[ \sum_t r(s_t, a_t) \right]}_{\text{(1) DARC Error Bound in Source}}$$

$$+ \|r\|_\mathcal{D} \Big[ \hat{\epsilon} + \underbrace{\inf_\zeta d_\mathcal{D}(\hat{\tau}^{src}_{\pi_{DARC}}, \hat{\tau}^{trg}_{\hat{\zeta}})}_{\text{(2.1) Approximation Error}} + \underbrace{2\hat{\mathcal{R}}^{(m)}_{\tau^{trg}_{\pi_{DARC}}} + 2W\hat{\mathcal{R}}^{(m)}_{\tau^{trg}_{\hat{\zeta}}} + (6W + 1)\Delta\sqrt{\log(4/\delta)/2m}}_{\text{(2.2) Estimation Error}} \Big].$$

$$\underbrace{\phantom{+ \|r\|_\mathcal{D} \Big[ \hat{\epsilon} + \inf_\zeta d_\mathcal{D}(\hat{\tau}^{src}_{\pi_{DARC}}, \hat{\tau}^{trg}_{\hat{\zeta}}) + 2\hat{\mathcal{R}}^{(m)}_{\tau^{trg}_{\pi_{DARC}}} + 2W\hat{\mathcal{R}}^{(m)}_{\tau^{trg}_{\hat{\zeta}}} + (6W + 1)\Delta\sqrt{\log(4/\delta)/2m} \Big]}}_{\text{(2) Imitation Learning Error Bound}}$$

**Remark 4.2.** *Our error bound depends on (1) the DARC error bound in the source domain and (2) the imitation learning generalization error, where (2) is further decomposed into (2.1) approximation error and (2.2) estimation error. This bound demonstrates how the two important components in our proposed approach contribute to a good performance. Firstly, we would want a well-trained policy on the source to reduce (1), which can be achieved by a good policy learning algorithm and well-trained classifiers for reward modification. Secondly, we utilize imitation learning from observation to transfer the experience to the source. (2.1) depends on the upper bound of the importance weight, which can be decreased with a richer policy class or when the dynamics shift becomes smaller. Additionally, a better imitation can be also achieved by increasing the complexity of the discriminator function class and the number of samples, which pushes (2.2) to be smaller.*

### 4.1 Comparison with the Analysis of DARC

As we discussed in Section 3.1, the DARC algorithm [3] trains a policy $\pi_{\text{DARC}}$ on the source domain via matching the distribution of trajectories generated by $\pi_{\text{DARC}}$ in the source and the distribution of the optimal trajectory in the target domain. Consequently, the learned policy $\pi_{\text{DARC}}$ will be suboptimal if it is directly deployed in the target domain.

In the DARC analysis, it is assumed that the optimal policy for the target domain $\pi^*$ lies in the *no exploit set* defined as follows [3, Assumption 1].

$$\Pi_{\text{no exploit}} \triangleq \left\{ \mathbb{E}_{a \sim \pi(a|s)} \left[ \sum_t \mathcal{D}_{\text{KL}}(p_{\text{src}}(s_{t+1}|s_t, a_t) || p_{\text{trg}}(s_{t+1}|s_t, a_t)) \right] \leq \epsilon \right\}. \tag{4.1}$$

Here, the *no exploit set* means that the experiences for any policy in this set are similar in the source and target domains. Consequently, any two policies in this *no exploit set* also receive similar expected rewards in the two domains, and thus the reward received by $\pi^*$ in the target domain is similar to

that received by $\pi_{\text{DARC}}$ in the target domain. Further, the objective function Eq. (3.3) of DARC is equivalent to the following constrained optimization.

$$\max_{\pi \in \Pi_{\text{no exploit}}} \mathbb{E}_{p_{\text{src}}, \pi} \left[ \sum_t r(s_t, a_t) + \mathcal{H}[a_t|s_t] \right]. \tag{4.2}$$

Thus, deploying the policy $\pi_{\text{DARC}}$ will not receive a huge performance degradation. However, the assumption that $\pi^* \in \Pi_{\text{no exploit}}$ is stringent and might not always be satisfied when the dynamics shift is large. When this assumption is violated, $\pi^*$ is not a good policy in the source domain, though it is the optimal policy in the target domain. Thus, the DARC policy which only optimizes the modified reward in the source domain will have significant performance degradation, as we have empirically shown in Figure 1 (a) and Figure 9. We also demonstrate this performance gap in Lemma A.1 in Appendix A when their assumption is not satisfied.

In contrast, our algorithm DARAIL does not assume the performance of $\pi_{\text{DARC}}$ in the source domain to be close to the performance of $\pi^*$ in the target domain. Instead, we only assume that the importance weight is somehow bounded, meaning that the dynamics shift is bounded. The error bound of our algorithm presented in Theorem 4.1 is controlled by imitation learning, which transfers the performance of $\pi_{\text{DARC}}$ in the source domain to that of $\pi^*$ in the target domain without assuming $\pi^* \in \Pi_{\text{no exploit}}$. Therefore, our algorithm can work well even in the cases shown in Figure 1 (a) and Figure 9 where the experience of $\pi_{\text{DARC}}$ is very distinctive in the source and target domains.

## 5   Experiment

In this section, we conduct experiments on off-dynamics reinforcement learning settings on four OpenAI environments: *HalfCheetah-v2*, *Ant-v2*, *Walker2d-v2*, and *Reacher-v2*. We compare our method with seven baselines and demonstrate the superiority of the proposed DARAIL.

### 5.1   Experiments Setup

**Dynamics Shifts:** We examine our algorithm with two types of dynamics shift. **1) Broken environment.** Following previous work [3], we freeze the 0-index value to 0 in action: zero torque is applied to this joint, regardless of the commanded torque. Different from DARC [3], who only test their method in intact source and broken target environment, we further test our algorithm in the broken source and intact target environment, where the source has less support than the target domain. As discussed in Section 4.1, violating the $\pi^* \in \Pi_{\text{no exploit}}$ assumption leads to significant performance degradation for DARC and similar methods. When the source domain is intact, this assumption is more likely to hold and DARC can achieve a near-optimal policy in the target domain. So, besides the setting in DARC, we focus on a harder problem for off-dynamics RL where DARC is prone to failure due to the violation of the assumptions in Section 4.1. Further, for the Ant and Walker2d, the source environment is broken with $p_f = 0.8$ probability, which means that with 0.8 probability, the 0-index will be set to be 0, and 0.2 probability remains the original value. More details about the broken environment will be introduced in the Appendix C.3. **2) Modify parameters of the environment.** Besides the broken environment, we create dynamics shifts by modifying MuJoCo's configuration files for the target domain. Specifically, we modify one of the coefficients of {*gravity*, *density*} from 1.0 to one of the value $\{0.5, 1.5\}$.

**Baselines:** We first compare our method with DARC performance in the source and target domains. **DARC Training** and **DARC Evaluation**, defined as $\mathbb{E}_{p_{\text{src}}, \pi_{\text{DARC}}}[\sum_t r(s_t, a_t)]$ and $\mathbb{E}_{p_{\text{trg}}, \pi_{\text{DARC}}}[\sum_t r(s_t, a_t)]$ respectively, represent DARC performance in the two domains. We compare DARAIL with DARC training performance as we mimic the DARC behavior in the source domain, which should receive a similar reward as the DARC training reward in the source domain. We compare with DARC Evaluation to show that our method mitigates the problem of DARC and outperforms DARC in the target domain. Further, we compare our method DARAIL with several baselines that we describe as follows. *Importance Sampling for Reward* (**IS-R**) re-weights the reward in the transition with $\frac{p_{\text{trg}}(s_{t+1}|s_t, a_t)}{p_{\text{src}}(s_{t+1}|s_t, a_t)}$, and update the policy with reward $\frac{p_{\text{trg}}(s_{t+1}|s_t, a_t)}{p_{\text{src}}(s_{t+1}|s_t, a_t)} r(s_t, a_t)$ [18]. *Importance Sampling for SAC Actor and Critic Loss* (**IS-ACL**) [18] re-weights the transitions in the SAC actor and critic loss. **DAIL** is a reduction of DARAIL without reward augmentation. Model-based RL method **MBPO** [19] uses short model rollouts branched from real data to reduce the compounding errors of inaccurate models and decouple the model horizon from the task horizon. **MATL** [20] uses different modified rewards and is similar to our problem setting, except that they have access to rewards in

the target domain. Finally, we compare with generative adversarial reinforced action transformation (**GARAT**) [10], a grounded action transformation method that uses imitation learning to modify the action that is executed in the source domain to simulate the target transitions. More details of the baselines are in Appendix C.2.

**Experimental Details:** We perform weight clipping to all methods that use the importance weight, including the DARAIL, DAIL, IS-R, and IS-ACL, and select the $[0.01, 100]$ as the clipping interval for fair comparison, which works well for all methods. We also show that DARAIL is less sensitive to the importance of weight clipping in the next section. We conduct fair parameter tuning for our method and baselines, including learning rate, Gaussian noise scale, and learning frequency of the importance weight. We also tune the parameter for the imitation learning component in DARAIL and DAIL and notice that the higher update frequency tends to perform better, and experiment results are in Appendix D.2. More details are in Appendix D.4.

## 5.2 Results

We show the results of DARAIL and DARC in Table 1 and 2 for broken source and 1.5 gravity setting, respectively. And the results of other baselines are in Table 3 and 4. We refer to the results on other settings in the Appendix, including the intact source and broken target environment setting and the modification of different scales of the parameters in the configuration file. We will also empirically discuss why DARC works well in the broken target setting while fails in the broken source setting in Appendix C.6.

Table 1: Comparison of DARAIL with DARC, broken source environment.

|  | DARC Evaluation | DARC Training | Optimal in Target | DARAIL |
| --- | --- | --- | --- | --- |
| HalfCheetah | $4133 \pm 828$ | $6995 \pm 30$ | $8543 \pm 230$ | $7067 \pm 176$ |
| Ant | $4280 \pm 33$ | $5197 \pm 155$ | $6183 \pm 348$ | $5357 \pm 79$ |
| Walker2d | $2669 \pm 788$ | $3896 \pm 523$ | $3899 \pm 214$ | $4366 \pm 434$ |
| Reacher | $-26.3 \pm 3.3$ | $-11.2 \pm 2.9$ | -7.2 $\pm$ 1.2 | $-13.7 \pm 0.9$ |

Table 2: Comparison of DARAIL with DARC, 1.5 gravity.

|  | DARC Evaluation | DARC Training | Optimal in Target | DARAIL |
| --- | --- | --- | --- | --- |
| HalfCheetah | $653 \pm 142$ | $4897 \pm 653$ | $6894 \pm 491$ | $4093 \pm 1021$ |
| Ant | $1587 \pm 594$ | $2170 \pm 258$ | $5320 \pm 429$ | $3472 \pm 771$ |
| Walker2d | $257 \pm 28$ | $4130 \pm 689$ | $4254 \pm 345$ | $4409 \pm 401$ |
| Reacher | $-55.3 \pm 10.3$ | $-17.2 \pm 3.8$ | -8.3 $\pm$ 1.3 | $-9.5 \pm 0.22$ |

**The Suboptimality of DARC and DARAIL outperforms DARC** By comparing DARC Training and DARC Evaluation in Table 1 and 2 we demonstrate that there is a performance degradation of $\pi_{\text{DARC}}$ deployed in the target domain on all four environments. $\pi_{\text{DARC}}$ reward in the target domain is about $40\%$ lower than $\pi_{\text{DARC}}$ reward in the source domain on average for broken source setting, and the degradation can be more severe in the changing gravity and density setting. Also, $\pi_{\text{DARC}}$ reward in the target domain is significantly lower than the target optimal reward. The training reward curves of DARC of the broken source environment setting are in Appendix C.5, clearly showing performance degradation when deployed in the target domain. Further, DARAIL outperforms the DARC evaluation performance.

**DARAIL Outperforms Baselines** We show the result of DARAIL and baselines in Table 3, 4. The training curves of other settings are in Appendix C.4. In all four environments, DARAIL outperforms the $\pi_{\text{DARC}}$ reward in the target domain. DARAIL also achieves better performance or the same level of rewards compared to the $\pi_{\text{DARC}}$ in the source domain as shown in Table 1 and 2, which is our expert policy for the imitation step. Compared with the DAIL, DARAIL has a much better performance, which demonstrates the effectiveness of the reward estimator $R_{AE}$. Compared with the two important weighting methods, IS-R and IS-ACL, in broken source settings, DARAIL outperforms IS-R in four environments and IS-ACL in Ant and Walker2d. IS-ACL and DARAIL achieve similar rewards in HalfCheetah and Reacher. And in modifying configuration settings, DARAIL outperforms IS-R and IS-ACL. Our method outperforms MBPO, MATL, and GARAT in all environments.

**DARAIL is Less Sensitive to Extreme Values in Importance Weights** Though IS-ACL achieves comparable performance with DARAIL on some tasks shown in Table 3, it is highly sensitive to

Table 3: Comparison of DARAIL with baselines in off-dynamics RL, broken source environment.

|  | DAIL | IS-R | IS-ACL | MBPO | MATL | GARAT | DARAIL |
|---|---|---|---|---|---|---|---|
| HalfCheetah | $6402 \pm 362$ | $6007 \pm 863$ | $6934 \pm 231$ | $4323 \pm 7$ | $1538 \pm 616$ | $5877 \pm 382$ | $\mathbf{7067} \pm 176$ |
| Ant | $3239 \pm 395$ | $1463 \pm 1055$ | $2753 \pm 94$ | $2445 \pm 13$ | $2006 \pm 17$ | $3380 \pm 268$ | $\mathbf{5357} \pm 79$ |
| Walker2d | $2330 \pm 156$ | $3092 \pm 434$ | $3881 \pm 269$ | $1012 \pm 41$ | $250 \pm 5$ | $3296 \pm 284$ | $\mathbf{4366} \pm 434$ |
| Reacher | $-13.9 \pm 1.1$ | $-17.6 \pm 0.25$ | $-14.1 \pm 0.16$ | $-14.3 \pm 2$ | $-30 \pm 10$ | $-14.7 \pm 2.6$ | $\mathbf{-13.7} \pm 0.9$ |

Table 4: Comparison of DARAIL with baselines in off-dynamics RL, 1.5 gravity.

|  | DAIL | IS-R | IS-ACL | MBPO | MATL | GARAT | DARAIL |
|---|---|---|---|---|---|---|---|
| HalfCheetah | $2666 \pm 2037$ | $2718 \pm 1978$ | $3576 \pm 312$ | $619 \pm 311$ | $337 \pm 205$ | $3825 \pm 437$ | $\mathbf{4093} \pm 1021$ |
| Ant | $990 \pm 251$ | $1712 \pm 393$ | $2396 \pm 573$ | $989 \pm 13$ | $1376 \pm 466$ | $1961 \pm 115$ | $\mathbf{3472} \pm 771$ |
| Walker2d | $525 \pm 142$ | $1543 \pm 604$ | $1369 \pm 705$ | $870 \pm 451$ | $1419 \pm 489$ | $630 \pm 230$ | $\mathbf{4409} \pm 401$ |
| Reacher | $-16.5 \pm 1.1$ | $-14.6 \pm 0.8$ | $-47.4 \pm 8.3$ | $-18.3 \pm 0.9$ | $-17.6 \pm 0.7$ | $-16.7 \pm 0.3$ | $\mathbf{-9.5} \pm 0.22$ |

the clipping interval of importance weight. In Figure 2, we show the performance of DARAIL and IPS-ACL on different importance weight clipping intervals in the broken source setting, and DARAIL outperforms IPS-ACL on all tasks. If the clipping interval is too large, IPS-ACL suffers from high variance, thus harming the performance. If the clipping interval is too small, the effective information about the dynamics shift is lost. On the other hand, DARAIL is less sensitive to it, which is an inherent property of our $R_{AE}$. Furthermore, in Figure 2, for IPS-ACL, the training curve for $[0.001, 1000]$ clipping interval has a much larger variance than $[0.1, 10]$ clipping interval, while our method does not suffer from such a high variance. This also demonstrates that our proposed reward estimator $R_{AE}$ is a more robust estimator and less affected by the importance weight.

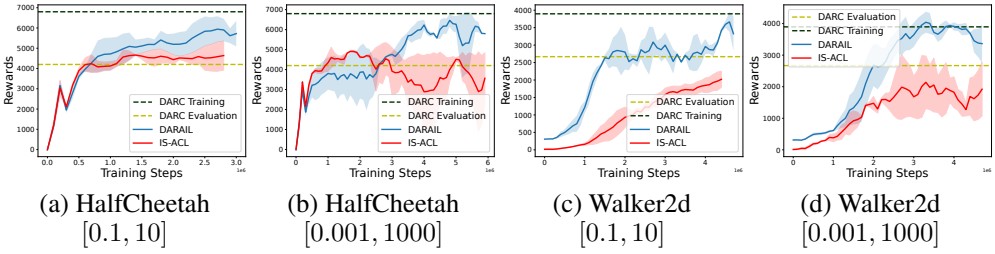

| (a) HalfCheetah $[0.1, 10]$ | (b) HalfCheetah $[0.001, 1000]$ | (c) Walker2d $[0.1, 10]$ | (d) Walker2d $[0.001, 1000]$ |
|---|---|---|---|

Figure 2: Performance of DARAIL and IPS-ACL on HalfCheetah and Walker2d under different importance weight clipping intervals. DARAIL outperforms IPS-ACL on all tasks. In Table 3, IPS-ACL receives comparable performance with DARAIL with the clipping interval [0.01,100], while the performance decreases significantly with different intervals.

**DARAIL's Performance on Different Magnitudes of Shifts** In our broken action environments, as we create the off-dynamics shift by (probabilistically) freezing one action dimension in the source domain, we can control the off-dynamics shift magnitudes by controlling the broken probability. For the same environment, the larger the $p_f$ is, the higher the probability of freezing the 0-index action, thus a larger dynamics shift. We consider $p_f = [0.2, 0.5, 0.8]$ for Ant, respectively and the experiment results is shown in Figure 3. From left to right, as the dynamics shift increases, we observe that the DARC performance decreases, and DARAIL outperforms DARC on all tasks.

## 6 Related Work

**Off-dynamics RL** Off-dynamics RL [3] is a specific domain adaptation [21, 22] and transfer learning problem in the RL domain [23] where the goal is to learn a policy from a source domain to adapt to a target domain where the dynamics are different. Similar to many works in off-policy evaluation (OPE) [12] in bandit and offline/off-policy RL [13, 24], an importance weight approach can be used to account for the difference between the transition dynamics with $\frac{p_{\mathrm{trg}}(s_{t+1}|s_t,a_t)}{p_{\mathrm{src}}(s_{t+1}|s_t,a_t)}$. However, this method can easily suffer from high variance due to the estimation bias of $p_{\mathrm{src}}(s_{t+1}|s_t, a_t)$ [12]. Another line of method for the off-dynamics RL is through reward shaping [3, 5]. DARC [3] learns a policy from a modified reward function that accounts for the dynamics shifts through a trajectories distribution matching objective. [6] proposed an unsupervised domain adaptation method with KL regularized objective, which uses the same reward modification techniques trajectories distribution matching

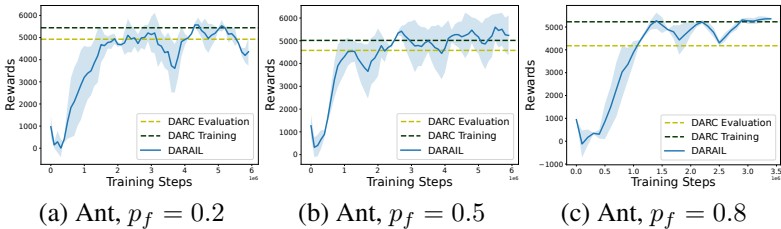

(a) Ant, $p_f = 0.2$  (b) Ant, $p_f = 0.5$  (c) Ant, $p_f = 0.8$

Figure 3: Performance of DARC and DARAIL under different off-dynamics shifts on Ant. Action 0 is frozen (set to be 0) with probability $p_f$ in the source domain. From left to right, the off-dynamics shift becomes larger. As the shift becomes larger, the gap between DARC Training and DARC Evaluation is larger. Our method outperforms DARC on different dynamics shift.

objective in DARC [3]. These reward-shaping methods all face the same problem: they will not recover the optimal policy in the target domain and will suffer from performance degradation in the target domain, but the policy's experience in the source domain is similar to the optimal policy in the target domain. Similarly, [25] proposes a state-regularized policy optimization method that constrains the state distribution to be similar in the source and target domain by adding a constraint term in the reward. However, this will also lead to suboptimal policy in the target domain like DARC. Different from DARC, Mutual Alignment Transfer Learning (MATL) [20] uses different modified rewards with GAN [26] to align the trajectories generated in the source and the target domain, but it requires access to the target domain reward. There is also work [27] that solves the off-dynamics RL problem by training a distributionally robust policy in the source domain by assuming that the target domain's transition probability is in an ambiguity set defined around the transition probability of the source domain. Our method builds on DARC, inspired by its property in the source domain, overcoming the issues in DARC and similar methods by mimicking the $\pi_{\text{DARC}}$ behavior in the source domain.

**Imitation Learning** Imitation learning (IL) is another line of work that can be applied to off-dynamics problems by mimicking the expert demonstration in the target domain. Generative adversarial imitation learning, [7, 28–30, 8, 31, 32], frames IL as an occupancy-measure matching or divergence minimization problem, which minimizes the divergence of the generated trajectories and the expert demonstration. Building on GAN [26], it uses the RL algorithm as a generator and a classifier as a discriminator to achieve this. Imitation learning from observation (*Ifo*) [33–35] is recently proposed to mimic the expert's behavior without knowing which actions the expert took. In the off-dynamics RL setting, recent work on IL under dynamics mismatch [11, 10, 36] can transfer a policy learned in the source to the target domain with minimal interaction with the target domain. However, these methods require high-quality and sufficient expert demonstrations and also the expert demonstrations might not be the optimal trajectories for the target domain, resulting in a suboptimal policy. Our method, DARAIL, transfers the DARC policy's behavior in the source to the target domain through imitation learning from observation so that the new policy will behave like the optimal policy in the target domain. Furthermore, we propose a novel and practical reward estimator with the signal from the discriminator and the reward from the source domain for the policy optimization.

# 7 Conclusion

In this paper, we propose Domain Adaptation and Reward Augmented Imitation Learning (DARAIL) for off-dynamics RL. We recognize the drawbacks of DARC and its following work with the same modified rewards function. We demonstrate that DARC or similar reward modification methods can only obtain a near-optimal policy in the target domain. We then propose to mimic the trajectory distribution generated by DARC in the source domain. Specifically, we propose a reward-augmented estimator for the policy optimization step in imitation learning from observation. Theoretically, we established the finite sample upper bounds of rewards for the proposed method, relaxing the restrictive assumption about the optimal policy in the previous work. Empirically, we conducted experiments on four Mujoco environments, demonstrating the superiority of our method. From the safety perspective, our method avoids directly training a policy in a high-risk environment. Our future work includes investigating off-dynamics reinforcement learning under safety constraints and more severe domain gaps in reinforcement learning.

## Acknowledgments

We would like to thank the anonymous reviewers for their helpful comments. YG was supported by the Center for Digital Health and Artificial Intelligence (CDHAI) of the Johns Hopkins University. PX was supported in part by the National Science Foundation (DMS-2323112) and the Whitehead Scholars Program at the Duke University School of Medicine. AL was partially supported by the Amazon Research Award, the Discovery Award of the Johns Hopkins University, and a seed grant from the JHU Institute of Assured Autonomy. The views and conclusions in this paper are those of the authors and should not be interpreted as representing any funding agency.

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

# A  Analysis of DARC

## A.1  DARC Objective

Figure 4 shows the objective of DARC, which minimizes the reverse KL divergence of the trajectories generated by the $\pi_{\text{DARC}}$ in the source domain and $\pi^*$ in the target domain. Note that the optimal policy is assumed to be proportional to the exponential form of the reward, i.e. $\pi^* \propto \exp(r(s_t, a_t))$. Given this assumption, the reverse KL divergence can be re-formulated to Eq. (3.3) with modified reward. So, the $\pi_{\text{DARC}}$ will not be optimal in the target domain but can generate trajectories in the source domain that resemble the optimal trajectories given the objective.

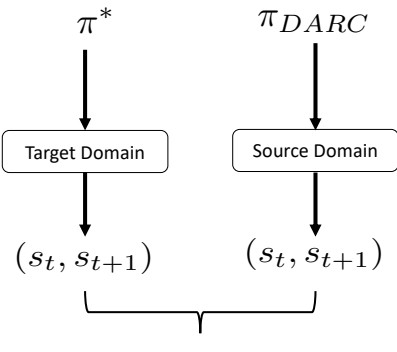

Figure 4: Optimization objective of DARC. DARC minimizes the reverse KL divergence of the trajectories generated by the $\pi_{\text{DARC}}$ and optimal policy $\pi^*$.

## A.2  DARC Error Bound

Now, we show that without the assumption of $\pi^* \in \Pi_{no\ exploit}$ in [3], the error of $\pi_{\text{DARC}}$ cannot be trivially bounded.

**Lemma A.1.** *If $\pi^* \notin \Pi_{no\ exploit}$, the error bound of the $\pi_{DARC}$ is in the following form:*

$$\mathrm{E}_{p_{trg}, \pi^*}\left[\sum_t r(s_t, a_t) + \mathcal{H}[a_t|s_t]\right] - \mathrm{E}_{p_{trg}, \pi_{DARC}}\left[\sum_t r(s_t, a_t) + \mathcal{H}[a_t|s_t]\right]$$

$$\leq 2R_{max}\sqrt{\frac{1}{2}D_{KL}(p_{trg,\pi^*}(\tau), p_{src,\pi^*}(\tau))} + \sum_t TV(\pi_{DARC}(\cdot|s_t), \pi^*(\cdot|s_t)) \max_{s_t, a_t, s_{t+1}} \Delta r(s_t, a_t, s_{t+1})$$

$$+ 2R_{max}\sqrt{\epsilon/2}.$$

*Proof.* In [3] Lemma B.2, they show that for any policy $\pi \in \Pi_{no\ exploit}$, the following inequality holds:

$$\left|\mathrm{E}_{p_{src}, \pi}\left[\sum_t r(s_t, a_t) + \mathcal{H}_\pi[a_t|s_t]\right] - \mathrm{E}_{p_{trg}, \pi}\left[\sum_t r(s_t, a_t) + \mathcal{H}_\pi[a_t|s_t]\right]\right| \leq 2R_{max}\sqrt{\epsilon/2}, \quad (A.1)$$

where $R_{max}$ refers to the maximum entropy-regularized return of any trajectories. However, the inequality Eq. (A.1) only holds for $\pi_{\text{DARC}}$, not for $\pi^*$. Now, we show that without the assumption $\pi^* \in \Pi_{no\ exploit}$, the error could not be bounded trivially.

We start with the same decomposition. Therefore, we have

$$\mathrm{E}_{p_{trg}, \pi^*}\left[\sum_t r(s_t, a_t) + \mathcal{H}[a_t|s_t]\right] - \mathrm{E}_{p_{trg}, \pi_{DARC}}\left[\sum_t r(s_t, a_t) + \mathcal{H}[a_t|s_t]\right]$$

$$= \mathrm{E}_{p_{\mathrm{trg}},\pi^*}\left[\sum_t r(s_t,a_t) + \mathcal{H}[a_t|s_t]\right] - \mathrm{E}_{p_{\mathrm{src}},\pi^*}\left[\sum_t r(s_t,a_t) + \mathcal{H}[a_t|s_t]\right]$$

$$\underbrace{\phantom{= \mathrm{E}_{p_{\mathrm{trg}},\pi^*}\left[\sum_t r(s_t,a_t) + \mathcal{H}[a_t|s_t]\right] - \mathrm{E}_{p_{\mathrm{src}},\pi^*}\left[\sum_t r(s_t,a_t) + \mathcal{H}[a_t|s_t]\right]}}_{I_1}$$

$$+ \mathrm{E}_{p_{\mathrm{src}},\pi^*}\left[\sum_t r(s_t,a_t) + \mathcal{H}[a_t|s_t]\right] - \mathrm{E}_{p_{\mathrm{src}},\pi_{\mathrm{DARC}}}\left[\sum_t r(s_t,a_t) + H_{\pi^*}[a_t|s_t]\right]$$

$$\underbrace{\phantom{xxxxxxxxxxxxxxxxxxxxxxxxxxxxxxxxxxxxxxxxxxxxxxxxxxxxxxxxxxxxxxxxxx}}_{I_2}$$

$$+ \mathrm{E}_{p_{\mathrm{src}},\pi_{\mathrm{DARC}}}\left[\sum_t r(s_t,a_t) + \mathcal{H}[a_t|s_t]\right] - \mathrm{E}_{p_{\mathrm{trg}},\pi_{\mathrm{DARC}}}\left[\sum_t r(s_t,a_t) + \mathcal{H}[a_t|s_t]\right]. \quad (A.2)$$

$$\underbrace{\phantom{xxxxxxxxxxxxxxxxxxxxxxxxxxxxxxxxxxxxxxxxxxxxxxxxxxxxxxxxxxxxxxxxxx}}_{I_3}$$

In the original proof of [3], they bound the three terms based on the following idea:

For the term $I_1$, they directly assume $\pi^* \in \Pi_{no\ exploit}$ and obtain $I_1 \le 2R_{max}\sqrt{\epsilon/2}$ based on inequality Eq. (A.1). However, without the $\pi^* \in \Pi_{no\ exploit}$, the upper bound is not valid. A valid upper bound should be:

$$I_1 = \mathrm{E}_{p_{\mathrm{trg}},\pi^*}\left[\sum_t r(s_t,a_t) + \mathcal{H}[a_t|s_t]\right] - \mathrm{E}_{p_{\mathrm{src}},\pi^*}\left[\sum_t r(s_t,a_t) + \mathcal{H}[a_t|s_t]\right]$$

$$= \sum_\tau (p_{\mathrm{trg},\pi^*}(\tau) - p_{\mathrm{src},\pi^*}(\tau))\left[\sum_t r(s_t,a_t) + \mathcal{H}[a_t|s_t]\right]$$

$$\le R_{max}\|p_{\mathrm{trg},\pi^*}(\tau) - p_{\mathrm{src},\pi^*}(\tau)\|_\infty$$

$$\le 2R_{max}\sqrt{\frac{1}{2}D_{KL}(p_{\mathrm{trg},\pi^*}(\tau), p_{\mathrm{src},\pi^*}(\tau))}. \quad (A.3)$$

If $\pi^* \in \Pi_{no\ exploit}$ holds, we have $D_{KL}(p_{\mathrm{trg},\pi^*}(\tau), p_{\mathrm{src},\pi^*}(\tau)) \le \epsilon$, which recovers the inequality Eq. (A.1). If it doesn't, we cannot trivially bound the $D_{KL}(p_{\mathrm{trg},\pi^*}(\tau), p_{\mathrm{src},\pi^*}(\tau))$.

For the term $I_2$, in the proof of [3], they also assume $\pi^* \in \Pi_{no\ exploit}$ and obtain the $I_2 \le 0$ based on the objective $\pi_{\mathrm{DARC}}$ maximizes the reward in the source domain with $\pi_{\mathrm{DARC}} \in \Pi_{no\ exploit}$. If $\pi^* \in \Pi_{no\ exploit}$ doesn't hold, we can bound this term by the following inequality:

$$\mathrm{E}_{p_{\mathrm{src}},\pi_{\mathrm{DARC}}}\left[\sum_t r(s_t,a_t) + \Delta r(s_t,a_t,s_{t+1}) + \mathcal{H}[a_t|s_t]\right]$$

$$\ge \mathrm{E}_{p_{\mathrm{src}},\pi^*}\left[\sum_t r(s_t,a_t) + \Delta r(s_t,a_t,s_{t+1}) + \mathcal{H}[a_t|s_t]\right],$$

which is equivalent to

$$\mathrm{E}_{p_{\mathrm{src}},\pi^*}\left[\sum_t r(s_t,a_t) + \mathcal{H}[a_t|s_t]\right] - \mathrm{E}_{p_{\mathrm{src}},\pi_{\mathrm{DARC}}}\left[\sum_t r(s_t,a_t) + \mathcal{H}[a_t|s_t]\right]$$

$$\le \mathrm{E}_{p_{\mathrm{src}},\pi^*}\left[\sum_t \Delta r(s_t,a_t,s_{t+1})\right] - \mathrm{E}_{p_{\mathrm{src}},\pi_{\mathrm{DARC}}}\left[\sum_t \Delta r(s_t,a_t,s_{t+1})\right] \quad (A.4)$$

$$\le \sum_t TV(\pi_{\mathrm{DARC}}(\cdot|s_t), \pi^*(\cdot|s_t)) \max_{s_t,a_t,s_{t+1}} \Delta r(s_t,a_t,s_{t+1}). \quad (A.5)$$

And the total variation of the two policies cannot be trivially bound as well. For the term $I_3$, we can easily bound it by applying the inequality Eq. (A.1) as $\pi_{\mathrm{DARC}} \in \Pi_{no\ exploit}$.

In summary, the bound without assuming $\pi^* \in \Pi_{no\ exploit}$ will be:

$$\mathrm{E}_{p_{\mathrm{trg}},\pi^*}\left[\sum_t r(s_t,a_t) + H[a_t|s_t]\right] - \mathrm{E}_{p_{\mathrm{trg}},\pi_{\mathrm{DARC}}}\left[\sum_t r(s_t,a_t) + \mathcal{H}[a_t|s_t]\right]$$

$$\leq 2R_{max}\sqrt{\frac{1}{2}D_{KL}(p_{\text{trg},\pi^*}(\tau), p_{\text{src},\pi^*}(\tau))} + \sum_t TV(\pi_{\text{DARC}}(\cdot|s_t), \pi^*(\cdot|s_t)) \max_{s_t,a_t,s_{t+1}} \Delta r(s_t, a_t, s_{t+1})$$

$$+ 2R_{max}\sqrt{\epsilon/2}.$$

This completes the proof. $\square$

# B  Theoretical Analysis of DARAIL

In this section, we prove our theoretical results.

**Definition B.1.** *(Neural Network Distance [37, 38]) For a class of neural networks $\mathcal{D}$, the neural network distance between two state-next state distributions, $\tau^{src}_{\pi_{DARC}}$ and $\tau^{trg}_{\zeta}$, is defined as*

$$d_{\mathcal{D}}(\tau^{src}_{\pi_{DARC}}, \tau^{trg}_{\zeta}) = \sup_{D \in \mathcal{D}} \left\{ \mathbb{E}_{(s_t,s_{t+1}) \sim \tau^{src}_{\pi_{DARC}}}[D(s_t, s_{t+1})] - \mathbb{E}_{(s_t,s_{t+1}) \sim \tau^{trg}_{\zeta}}[D(s_t, s_{t+1})] \right\}$$

$$= \sup_{D \in \mathcal{D}} \left\{ \mathbb{E}_{(s_t,s_{t+1}) \sim \tau^{src}_{\pi_{DARC}}}[D(s_t, s_{t+1})] - \mathbb{E}_{(s_t,s_{t+1}) \sim \tau^{src}_{\zeta}}[\rho(s_t, s_{t+1})D(s_t, s_{t+1})] \right\}.$$

**Definition B.2.** *(Empirical Rademacher Complexity) Given a function class $\mathcal{F}$, a dataset $X = (x_1, x_2, ..., x_n)$, i.i.d drawn from distribution $\mu$ and random variable $\sigma$ defined as $P(\sigma = 1) = P(\sigma = -1) = \frac{1}{2}$, the empirical Rademacher complexity is given by:*

$$\hat{\mathcal{R}}^{(n)}_{\mu} = \mathbb{E}_{\sigma}[\sup_{f \in \mathcal{F}}]\frac{1}{n}\sum_{i=1}^{n}\sigma_i f(x_i). \tag{B.1}$$

**Definition B.3.** *(Linear Span of the Discriminator) Consider a span of the discriminator class: $span(D) = \{c_0 + \sum_i^k c_i D_i : c_0 \in \mathbb{R}, D_i \in \mathcal{D}, n \in \mathbb{N}\}$. Assuming the ground truth reward function $r$ lies in the $span(\mathcal{D})$, then the compatible coefficient is defined as:*

$$\|r\|_{\mathcal{D}} = \inf \left\{ \sum_i^k |c_i| : r = c_0 + \sum_i^k c_i D_i, c_0 \in \mathbb{R}, D_i \in \mathcal{D}, n \in \mathbb{N} \right\}. \tag{B.2}$$

The *compatible coefficient* represents the minimum number of functions in $\mathcal{D}$ required to the reward function $r$, which means the complexity of the reward function $r$.

**Lemma B.4.** *(GAIL Generalization). Let $\pi_{DARC}$ be the expert policy and $\hat{\zeta}$ be the solution of the imitation learning algorithm. Let discriminator class $\mathcal{D}$ be a $\Delta$-bounded function, i.e. $|D(s_t, s_{t+1})| \leq \Delta$. Suppose reward function $r$ lies in the span of the discriminator class. Given $d_{\mathcal{D}}(\hat{\tau}^{src}_{\pi_{DARC}}, \hat{\tau}^{trg}_{\zeta}) - \inf_{\zeta} d_{\mathcal{D}}(\hat{\tau}^{src}_{\pi_{DARC}}, \hat{\tau}^{trg}_{\zeta}) \leq \hat{\epsilon}$ (empirical neural network distance achieved by imitation learning), the importance weight $\rho(s, s_{t+1})$ is bounded by $W$, $m$ is the number of the expert data, then $\forall \delta \in (0, 1)$, with probability at least $1 - \delta$, we have*

$$\mathbb{E}_{p_{src},\pi_{DARC}}\left[\sum_t r(s_t, a_t)\right] - \mathbb{E}_{p_{trg},\hat{\zeta}}\left[\sum_t r(s_t, a_t)\right]$$

$$\leq \|r_{\mathcal{D}}\| \left[ \inf_{\zeta} d_{\mathcal{D}}(\hat{\tau}^{src}_{\pi_{DARC}}, \hat{\tau}^{trg}_{\zeta}) + 2\hat{\mathcal{R}}^{(m)}_{\tau^{src}_{\pi_{DARC}}} + 2W\hat{\mathcal{R}}^{(m)}_{\tau^{trg}_{\zeta}} + (6W+1)\Delta\sqrt{\frac{log(4/\delta)}{2m}} + \hat{\epsilon} \right].$$

*Proof.* Given $d_{\mathcal{D}}(\hat{\tau}^{\text{src}}_{\pi_{\text{DARC}}}, \hat{\tau}^{\text{trg}}_{\zeta}) - \inf_{\zeta} d_{\mathcal{D}}(\hat{\tau}^{\text{src}}_{\pi_{\text{DARC}}}, \hat{\tau}^{\text{trg}}_{\zeta}) \leq \hat{\epsilon}$, we can have

$$d_{\mathcal{D}}(\tau^{\text{src}}_{\pi_{\text{DARC}}}, \tau^{\text{trg}}_{\zeta}) \leq d_{\mathcal{D}}(\tau^{\text{src}}_{\pi_{\text{DARC}}}, \tau^{\text{trg}}_{\zeta}) - d_{\mathcal{D}}(\hat{\tau}^{\text{src}}_{\pi_{\text{DARC}}}, \hat{\tau}^{\text{trg}}_{\zeta}) + \inf_{\zeta} d_{\mathcal{D}}(\hat{\tau}^{\text{src}}_{\pi_{\text{DARC}}}, \hat{\tau}^{\text{trg}}_{\zeta}) + \hat{\epsilon}.$$

We prove that $d_{\mathcal{D}}(\tau^{\text{src}}_{\pi_{\text{DARC}}}, \tau^{\text{trg}}_{\zeta}) - d_{\mathcal{D}}(\hat{\tau}^{\text{src}}_{\pi_{\text{DARC}}}, \hat{\tau}^{\text{trg}}_{\zeta})$ has an upper bound.

$$d_{\mathcal{D}}(\tau^{\text{src}}_{\pi_{\text{DARC}}}, \tau^{\text{trg}}_{\zeta}) - d_{\mathcal{D}}(\hat{\tau}^{\text{src}}_{\pi_{\text{DARC}}}, \hat{\tau}^{\text{trg}}_{\zeta})$$

$$= \sup_{D \in \mathcal{D}} \left[ \mathbb{E}_{(s_t,s_{t+1}) \sim \tau^{\text{src}}_{\pi_{\text{DARC}}}}[D(s_t, s_{t+1})] - \mathbb{E}_{(s_t,s_{t+1}) \sim \tau^{\text{trg}}_{\zeta}}[D(s_t, s_{t+1})] \right]$$

$$- \sup_{D \in \mathcal{D}} \left[ \mathbb{E}_{(s_t, s_{t+1}) \sim \hat{\tau}^{\text{src}}_{\pi_{\text{DARC}}}} [D(s_t, s_{t+1})] - \mathbb{E}_{(s_t, s_{t+1}) \sim \hat{\tau}^{\text{trg}}_{\zeta}} [D(s_t, s_{t+1})] \right]$$

$$\leq \sup_{D \in \mathcal{D}} \left[ \mathbb{E}_{(s_t, s_{t+1}) \sim \tau^{\text{src}}_{\pi_{\text{DARC}}}} [D(s_t, s_{t+1})] - \mathbb{E}_{(s_t, s_{t+1}) \sim \hat{\tau}^{\text{src}}_{\pi_{\text{DARC}}}} [D(s_t, s_{t+1})] \right]$$

$$+ \sup_{D \in \mathcal{D}} \left[ \mathbb{E}_{(s_t, s_{t+1}) \sim \tau^{\text{trg}}_{\zeta}} [D(s_t, s_{t+1})] - \mathbb{E}_{(s_t, s_{t+1}) \sim \hat{\tau}^{\text{trg}}_{\zeta}} [D(s_t, s_{t+1})] \right]$$

$$= \sup_{D \in \mathcal{D}} \left[ \mathbb{E}_{(s_t, s_{t+1}) \sim \tau^{\text{src}}_{\pi_{\text{DARC}}}} [D(s_t, s_{t+1})] - \mathbb{E}_{(s_t, s_{t+1}) \sim \hat{\tau}^{\text{src}}_{\pi_{\text{DARC}}}} [D(s_t, s_{t+1})] \right]$$

$$+ \sup_{D \in \mathcal{D}} \left[ \mathbb{E}_{(s_t, s_{t+1}) \sim \tau^{\text{src}}_{\zeta}} [\rho(s_t, s_{t+1}) D(s_t, s_{t+1})] - \mathbb{E}_{(s_t, s_{t+1}) \sim \hat{\tau}^{\text{src}}_{\zeta}} [\rho(s_t, s_{t+1}) D(s_t, s_{t+1})] \right]$$

$$\leq \sup_{D \in \mathcal{D}} \left[ \mathbb{E}_{(s_t, s_{t+1}) \sim \tau^{\text{src}}_{\pi_{\text{DARC}}}} [D(s_t, s_{t+1})] - \mathbb{E}_{(s_t, s_{t+1}) \sim \hat{\tau}^{\text{src}}_{\pi_{\text{DARC}}}} [D(s_t, s_{t+1})] \right]$$

$$+ W \sup_{D \in \mathcal{D}} \left[ \mathbb{E}_{(s_t, s_{t+1}) \sim \tau^{\text{src}}_{\zeta}} [D(s_t, s_{t+1})] - \mathbb{E}_{(s_t, s_{t+1}) \sim \hat{\tau}^{\text{src}}_{\zeta}} [D(s_t, s_{t+1})] \right].$$

According to McDiarmid 's inequality, with probability at least $1 - \frac{\delta}{2}$, the following inequality holds

$$\sup_{D \in \mathcal{D}} \left[ \mathbb{E}_{(s_t, s_{t+1}) \sim \tau^{\text{src}}_{\pi_{\text{DARC}}}} [D(s_t, s_{t+1})] - \mathbb{E}_{(s_t, s_{t+1}) \sim \hat{\tau}^{\text{src}}_{\pi_{\text{DARC}}}} [D(s_t, s_{t+1})] \right]$$

$$\leq \mathbb{E} \left[ \sup_{D \in \mathcal{D}} | \mathbb{E}_{(s_t, s_{t+1}) \sim \tau \pi_{\text{DARC}}^{\text{src}}} [D(s_t, s_{t+1})] - \mathbb{E}_{(s_t, s_{t+1}) \sim \hat{\tau}^{\text{src}}_{\pi_{\text{DARC}}}} [D(s_t, s_{t+1})] | \right] + 2\Delta \sqrt{\frac{log(4/\delta)}{2m}}$$

$$\leq 2 \mathbb{E}_{\sigma, \tau^{\text{src}}_{\pi_{\text{DARC}}}} \left[ \sup_{D \in \mathcal{D}} \sum_{i=1}^{m} \frac{1}{m} \sigma_i D(s_i, s'_i) \right] + 2\Delta \sqrt{\frac{log(4/\delta)}{2m}}$$

$$\leq 2 \mathcal{R}^{(m)}_{\tau^{\text{src}}_{\pi_{\text{DARC}}}} + 2\Delta \sqrt{\frac{log(4/\delta)}{2m}}$$

$$\leq 2 \hat{\mathcal{R}}^{(m)}_{\tau^{\text{src}}_{\pi_{\text{DARC}}}} + 6\Delta \sqrt{\frac{log(4/\delta)}{2m}}.$$

By a similar derivation, we can have

$$W \sup_{D \in \mathcal{D}} \left[ \mathbb{E}_{(s_t, s_{t+1}) \sim \tau^{\text{src}}_{\zeta}} [D(s_t, s_{t+1})] - \mathbb{E}_{(s_t, s_{t+1}) \sim \hat{\tau}^{\text{src}}_{\zeta}} [D(s_t, s_{t+1})] \right]$$

$$\leq 2W \hat{\mathcal{R}}^{(m)}_{\tau^{\text{src}}_{\zeta}} + 6W\Delta \sqrt{\frac{log(4/\delta)}{2m}}.$$

Thus, we have

$$d_{\mathcal{D}}(\tau^{\text{src}}_{\pi_{\text{DARC}}}, \tau^{\text{trg}}_{\zeta}) - d_{\mathcal{D}}(\hat{\tau}^{\text{src}}_{\pi_{\text{DARC}}}, \hat{\tau}^{\text{trg}}_{\zeta})$$

$$\leq 2 \hat{\mathcal{R}}^{(m)}_{\tau^{\text{src}}_{\pi_{\text{DARC}}}} + 2W \hat{\mathcal{R}}^{(m)}_{\tau^{\text{trg}}_{\zeta}} + (6W + 1)\Delta \sqrt{\frac{log(4/\delta)}{2m}}.$$

Then, based on Theorem 2 in [38], we can conclude that

$$\mathbb{E}_{p_{\text{src}}, \pi_{\text{DARC}}} \left[ \sum_t r(s_t, a_t) \right] - \mathbb{E}_{p_{\text{trg}}, \hat{\zeta}} \left[ \sum_t r(s_t, a_t) \right]$$

$$\leq \|r_{\mathcal{D}}\| \left[ \inf_{\zeta} d_{\mathcal{D}}(\hat{\tau}^{\text{src}}_{\pi_{\text{DARC}}}, \hat{\tau}^{\text{trg}}_{\zeta}) + 2 \hat{\mathcal{R}}^{(m)}_{\tau^{\text{src}}_{\pi_{\text{DARC}}}} + 2W \hat{\mathcal{R}}^{(m)}_{\tau^{\text{src}}_{\zeta}} + (6W + 1)\Delta \sqrt{\frac{log(4/\delta)}{2m}} + \hat{\epsilon} \right].$$

This completes the proof. $\qquad \square$

**Theorem B.5.** *Let $\pi^* = \text{argmax}_\pi \mathbb{E}_{\pi, p_{trg}} [\sum_t r(s_t, a_t)]$ be the policy maximizing the cumulative reward in the target domain and $\hat{\zeta}$ be the policy learned from DARAIL. Let $m$ be the number of the expert demonstration and $\hat{\mathcal{R}}^{(m)}_\pi = \mathbb{E}_\sigma \left[ \sup_{D \in \mathcal{D}} \frac{1}{m} \sum_{i=1}^m \sigma_i D(s_t, s_{t+1}) \right]$ be the empirical Rademacher complexity. Let $B$ be the error bound of DARC in the source domain, i.e. $\mathbb{E}_{p_{src}, \pi^*_{DARC}} [\sum_t r(s_t, a_t) + \mathcal{H}[a_t|s_t]] - \mathbb{E}_{p_{src}, \pi_{DARC}} [\sum_t r(s_t, a_t)] \leq B$ and $W$ be the upper bound*

of the importance weight, i.e. $\rho(s_t, s_{t+1}) \leq W, \forall (s_t, s_{t+1})$. *Let discriminator class $\mathcal{D}$ be a $\Delta$-bounded function, i.e. $|D_\omega(s_t, s_{t+1})| \leq \Delta$ given any $(s_t, s_{t+1})$. $\|r\|_{\mathcal{D}}$ measures the richness of the discriminator to represent the ground truth reward as defined in Appendix B.2. $d_{\mathcal{D}}$ is a defined neural network distance between the $(s_t, s_{t+1})$ distributions generated by the $\pi_{DARC}$ and $\pi_{\hat{\zeta}}$ defined in Appendix B.1. Given the empirical training error of the imitation learning, i.e. $d_{\mathcal{D}}(\hat{\tau}^{src}_{\pi_{DARC}}, \hat{\tau}^{trg}_{\hat{\zeta}}) - \inf_\zeta d_{\mathcal{D}}(\hat{\tau}^{src}_{\pi_{DARC}}, \hat{\tau}^{trg}_\zeta) \leq \hat{\epsilon}, \forall \delta \in (0, 1)$, with probability at least $1 - \delta$, we have*

$$\mathbb{E}_{p_{trg}, \pi^*}\left[\sum_t r(s_t, a_t)\right] - \mathbb{E}_{p_{trg}, \hat{\zeta}}\left[\sum_t r(s_t, a_t)\right]$$

$$\leq \underbrace{\mathbb{E}_{p_{src}, \pi^*_{DARC}}\left[\sum_t r(s_t, a_t) + \mathcal{H}[a_t|s_t]\right] - \mathbb{E}_{p_{src}, \pi_{DARC}}\left[\sum_t r(s_t, a_t)\right]}_{\text{DARC Error Bound in Source}}$$

$$+ \underbrace{\|r\|_{\mathcal{D}}\left[\hat{\epsilon} + \underbrace{\inf_\zeta d_{\mathcal{D}}(\hat{\tau}^{src}_{\pi_{DARC}}, \hat{\tau}^{trg}_{\hat{\zeta}})}_{\text{Approximation Error}} + \underbrace{2\hat{\mathcal{R}}^{(m)}_{\tau^{trg}_{\pi_{DARC}}} + 2W\hat{\mathcal{R}}^{(m)}_{\tau^{trg}_{\hat{\zeta}}} + (6W+1)\Delta\sqrt{\frac{log(4/\delta)}{2m}}}_{\text{Estimation Error}}\right]}_{\text{Imitation Learning Error Bound}}.$$

*Proof.* We can first decompose it into three terms:

$$\mathbb{E}_{p_{trg}, \pi^*}\left[\sum_t r(s_t, a_t)\right] - \mathbb{E}_{p_{trg}, \hat{\zeta}}\left[\sum_t r(s_t, a_t)\right]$$

$$= \underbrace{\mathbb{E}_{p_{trg}, \pi^*}\left[\sum_t r(s_t, a_t)\right] - \mathbb{E}_{p_{src}, \pi^*_{DARC}}\left[\sum_t r(s_t, a_t) + \mathcal{H}(a_t|s_t)\right]}_{I_1}$$

$$+ \underbrace{\mathbb{E}_{p_{src}, \pi^*_{DARC}}\left[\sum_t r(s_t, a_t) + \mathcal{H}[a_t|s_t]\right] - \mathbb{E}_{p_{src}, \pi_{DARC}}\left[\sum_t r(s_t, a_t)\right]}_{I_2}$$

$$+ \underbrace{\mathbb{E}_{p_{src}, \pi_{DARC}}\left[\sum_t r(s_t, a_t)\right] - \mathbb{E}_{p_{trg}, \hat{\zeta}}\left[\sum_t r(s_t, a_t)\right]}_{I_3}.$$

Based on the formulation, $\pi^*_{DARC}$ can generate optimal trajectories for the target domain in the source domain so that $I_1 = 0$. Also, the $I_2$ term is the training error of the DARC and the entropy term of the optimal DARC policy, and we can assume together they are bounded by $B$. Then, we only need to bound the $I_3$ terms. Combining Lemma B.4, we have

$$\mathbb{E}_{p_{trg}, \pi^*}\left[\sum_t r(s_t, a_t)\right] - \mathbb{E}_{p_{trg}, \hat{\zeta}}\left[\sum_t r(s_t, a_t)\right]$$

$$\leq \underbrace{\mathbb{E}_{p_{src}, \pi^*_{DARC}}\left[\sum_t r(s_t, a_t) + \mathcal{H}[a_t|s_t]\right] - \mathbb{E}_{p_{src}, \pi_{DARC}}\left[\sum_t r(s_t, a_t)\right]}_{\text{DARC Error Bound in Source}}$$

$$+ \underbrace{\|r\|_{\mathcal{D}}\left[\hat{\epsilon} + \underbrace{\inf_\zeta d_{\mathcal{D}}(\hat{\tau}^{src}_{\pi_{DARC}}, \hat{\tau}^{trg}_{\hat{\zeta}})}_{\text{Approximation Error}} + \underbrace{2\hat{\mathcal{R}}^{(m)}_{\tau^{trg}_{\pi_{DARC}}} + 2W\hat{\mathcal{R}}^{(m)}_{\tau^{trg}_{\hat{\zeta}}} + (6W+1)\Delta\sqrt{\frac{log(4/\delta)}{2m}}}_{\text{Estimation Error}}\right]}_{\text{Imitation Learning Error Bound}}.$$

$\square$

# C  Additional Experimental Details and Results

Code is available at https://github.com/guoyihonggyh/Off-Dynamics-Reinforcement-Learning-via-Domain-Adaptation-and-Reward-Augmented-Imitation.

## C.1  Estimation of $\Delta r(s_t, a_t, s_{t+1})$ and importance weight $\frac{p_{\text{trg}}(s_{t+1}|s_t, a_t)}{p_{\text{src}}(s_{t+1}|s_t, a_t)}$

Following the DARC [3], the importance weight can be estimated with the following two binary classifiers $p(\text{trg}|s_t, a_t)$ and $p(\text{trg}|s_t, a_t, s_{t+1})$ with Bayes' rules:

$$p(\text{trg}|s_t, a_t, s_{t+1}) = p_{\text{trg}}(s_{t+1}|s_t, a_t)p(s_t, a_t|\text{trg})p(\text{trg})/p(s_t, a_t, s_{t+1}), \qquad (\text{C.1})$$

$$p(s_t, a_t|\text{trg}) = p(\text{trg}|s_t, a_t)p(s_t, a_t)/p(\text{trg}). \qquad (\text{C.2})$$

Replacing the $p(s_t, a_t|\text{trg})$ in Eq. (C.1) with Eq. (C.2), we obtain:

$$p_{\text{trg}}(s_{t+1}|s_t, a_t) = \frac{p(\text{trg}|s_t, a_t, s_{t+1})p(s_t, a_t, s_{t+1})}{p(\text{trg}|s_t, a_t)p(s_t, a_t)}.$$

Similarly, we can obtain the $p_{\text{src}}(s_{t+1}|s_t, a_t) = \frac{p(\text{src}|s_t, a_t, s_{t+1})p(s_t, a_t, s_{t+1})}{p(\text{src}|s_t, a_t)p(s_t, a_t)}$.

We can calculate the $\Delta r(s_t, a_t, s_{t+1})$ following:

$$
\begin{aligned}
\rho(s_t, s_{t+1}) &= \log\left(\frac{p_{\text{trg}}(s_{t+1}|s_t, a_t)}{p_{\text{src}}(s_{t+1}|s_t, a_t)}\right) \\
&= \log p(\text{trg}|s_t, a_t, s_{t+1}) - \log p(\text{trg}|s_t, a_t) + \log p(\text{src}|s_t, a_t, s_{t+1}) - \log p(\text{src}|s_t, a_t).
\end{aligned}
$$

$\rho(s_t, s_{t+1})$ can be obtained from $\rho(s_t, s_{t+1}) = \exp\left[\Delta r(s_t, a_t, s_{t+1})\right]$

**Training the classifier $p(\text{trg}|s_t, a_t)$ and $p(\text{trg}|s_t, a_t, s_{t+1})$** The two classifiers are parameterized bu $\theta_{\text{SA}}$ and $\theta_{\text{SAS}}$. To update the two classifiers, we sample one mini-batch of data from the source replay buffer $D_{\text{src}}^{\zeta}$ and the target replay buffer $D_{\text{src}}^{\zeta}$ respectively. Imbalanced data is considered here as each time we sample the same amount of data from the source and target domain buffer. Then, the parameters are learned by minimizing the standard cross-entropy loss:

$$\mathcal{L}_{\text{SAS}} = -\mathbb{E}_{\mathcal{D}_{\text{src}}^{\zeta}}\left[\log p_{\theta_{\text{SAS}}}(\text{trg}|s_t, a_t, s_{t+1})\right] - \mathbb{E}_{\mathcal{D}_{\text{trg}}^{\zeta}}\left[\log p_{\theta_{\text{SAS}}}(\text{trg}|s_t, a_t, s_{t+1})\right],$$

$$\mathcal{L}_{\text{SA}} = -\mathbb{E}_{\mathcal{D}_{\text{src}}^{\zeta}}\left[\log p_{\theta_{\text{SA}}}(\text{trg}|s_t, a_t, s_{t+1})\right] - \mathbb{E}_{\mathcal{D}_{\text{trg}}^{\zeta}}\left[\log p_{\theta_{\text{SA}}}(\text{trg}|s_t, a_t, s_{t+1})\right].$$

Thus, $\theta = (\theta_{\text{SAS}}, \theta_{\text{SA}})$ is obtained from:

$$
\begin{aligned}
\theta &= \underset{\theta}{\operatorname{argmin}} \, \mathcal{L}_{CE}(\mathcal{D}_{\text{src}}^{\zeta}, \mathcal{D}_{\text{trg}}^{\zeta}) \\
&= \underset{\theta}{\operatorname{argmin}}[\mathcal{L}_{\text{SAS}} + \mathcal{L}_{\text{SA}}]
\end{aligned}
$$

## C.2  Description of Baseline Methods

**Importance Sampling for Reward (IS-R)** With $(s_t, a_t, s_{t+1})$ from the source domain, the IS-R directly re-weight the reward in each transition. We can view IS-R as learning the SAC with rewards $\frac{p_{\text{trg}}(s_{t+1}|s_t, a_t)}{p_{\text{src}}(s_{t+1}|s_t, a_t)}r_t(s_t, a_t)$ and seeking to maximize the following objective:

$$\max_{\pi} \mathbb{E}_{(s_t, a_t, s_{t+1})\sim\pi(\cdot|s_t)\times p_{\text{src}}(\cdot|s_t, a_t)}\left[\sum_t \frac{p_{\text{trg}}(s_{t+1}|s_t, a_t)}{p_{\text{src}}(s_{t+1}|s_t, a_t)}r_t(s_t, a_t)\right].$$

**Importance Sampling for SAC Actor and Critic Loss (IS-ACL)** Another way of doing importance sampling is by re-weighting the actor and critic loss in SAC. The loss for the Q-network in SAC becomes:

$$\min_{\phi} \mathbb{E}_{(s_t, a_t, s_{t+1})\sim\pi(\cdot|s_t)\times p_{\text{src}}(\cdot|s_t, a_t)}\left[\frac{p_{\text{trg}}(s_{t+1}|s_t, a_t)}{p_{\text{src}}(s_{t+1}|s_t, a_t)}(Q_{\phi}(s_t, a_t) - y(s_t, a_t, d))^2\right]$$

where $d$ is the done signal, and the target is given by:

$$y(s_t, a_t, d) = r + \gamma(1 - d) \left[ \min_{j=1,2} Q_{\text{trg},j}(s_{t+1}, a') - \alpha \log \pi(a'|s_{t+1}) \right], a' \sim \pi(a|s_{t+1}).$$

The actor loss is:

$$\max_\pi \mathbb{E}_{a \sim \pi} \frac{p_{\text{trg}}(s_{t+1}|s_t, a_t)}{p_{\text{src}}(s_{t+1}|s_t, a_t)} \left[ Q^\pi(s, a) - \alpha \log \pi(a|s) \right].$$

**DAIL** In DARAIL, the policy is optimized with the reward estimator $R_{AE}$ with the true reward from the source domain. We also want to compare the vanilla imitation learning with importance weight. The objective is:

$$\min_\zeta \max_{D_\omega} \left\{ \mathbb{E}_{p_{\text{src}}, \zeta} \left[ \sum_t \rho(s_t, s_{t+1}) \log D_\omega(s_t, s_{t+1}) \right] + \mathbb{E}_{(s_t, s_{t+1}) \sim \tau_{\pi_{\text{DARC}}}^{\text{src}}} \left[ \sum_t \log(1 - D_\omega(s_t, s_{t+1})) \right] \right\},$$
(C.3)

Then, following the Eq.(C.3), the objective of policy optimization without the reward estimator is:

$$\max_\zeta \mathbb{E}_{p_{\text{src}}, \zeta} \left[ \sum_t -\rho(s_t, s_{t+1}) \log D_\omega(s_t, s_{t+1}) \right].$$
(C.4)

We can view it as a reduced version of our proposed method, which uses the reward function provided by the discriminator and importance weight.

**MBPO** [19]. MBPO is a model-based RL method. We train the MBPO in the source domain and deploy it to the target domain.

**MATL** [20]. MATL modified the reward on both the source and target domains and aligned the trajectories on both domains. Unlike our method, they need access to the reward from the target domain.

**GARAT**[10] GARAT is a grounded action transformation approach that simulates target transitions $(s_t, a_t, s_{t+1}, r)$ in the source domain with modified action, where the modified action is learned from imitation learning.

## C.3   Broken with probability $p_f$

As discussed, we use the *broken with probability* for Ant and Walker2d. The dynamics shift created by freezing one action varies across environments. For instance, in the Ant robot, the 0-index controls the rotor between the torso and front left hip, while in the HalfCheetah, the 0-index controls the back thigh rotor. So, the broken Ant experiences a larger shift than the broken HalfCheetah if we break the 0-index for both environments. Also, the broken environment in Walker2d and Ant creates such a large dynamics shift that it is overly difficult to adapt from the source domain, i.e., DARC cannot obtain the optimal reward in the source domain. We then introduce the *broken with probability $p_f$* to better control the magnitude of dynamics shift. *Broken with probability $p_f$* means the 0-index action is frozen with probability $p_f$ and follows the commanded torque with probability $1 - p_f$. In Reacher and HalfCheetah, the source environment is broken with probability 1. Ant and Walker2d's source domain is broken with a probability of 0.8.

Figure 5 shows the performance of DARC in Ant and Walker2d under different broken probability $p_f$ in the source domain. We can observe that when $p_f = 1.0$, the performance degradation of evaluating in the target domain is larger than the $p_f = 0.8$ case. Also, when $p_f = 1.0$, the DARC evaluation performance in the target domain is close to 0. Moreover, we notice that in the $p_f = 1.0$ case, DARC training performance in the source domain receives a much lower reward than the $p_f = 0.8$ case. However, we want to mimic the DARC behavior in the imitation learning, so we want DARC to be able to receive optimal reward in the source domain. Thus, for the Ant and Walker2d environment, we choose $p_f = 0.8$ for the source domain.

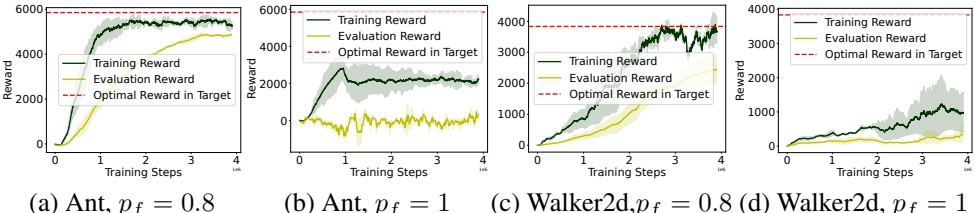

(a) Ant, $p_f = 0.8$      (b) Ant, $p_f = 1$      (c) Walker2d, $p_f = 0.8$ (d) Walker2d, $p_f = 1$

Figure 5: Training reward in the source domain, i.e. $\mathbb{E}_{\pi_{\mathrm{DARC}}, p_{\mathrm{src}}}[\sum_t r(s_t, a_t)]$, and evaluation reward in the target domain , i.e. $\mathbb{E}_{\pi_{\mathrm{DARC}}, p_{\mathrm{trg}}}[\sum_t r(s_t, a_t)]$, for DARC in Ant and Walker2d with different broken probability $p_f$ in the source domain. (a) and (c) shows the performance of DARC under $p_f = 0.8$, and (a) and (c) shows the performance of DARC under $p_f = 1.0$. The performance of DARC under $p_f = 1.0$ is much worse than the case $p_f = 0.8$, and the performance gap between DARC in the source and target is larger, showing that the dynamics shift is overly large to adapt and learn a good expert demonstration.

## C.4   Training Curve of the DARAIL and Baselines

We show the training curve of DARAIL and baselines in different environments under the broken source environment setting in Figure 6 corresponding to the result in Table 3. We also show the training curve of modifying the configuration in Figure 7 and 8.

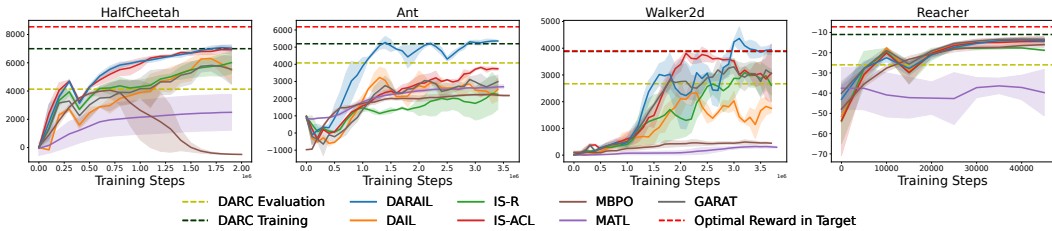

Figure 6: Upper horizon line: DARC reward in the source domain. Lower horizon line: DARC reward in the target domain. The figures show the mean value of multiple runs and the standard deviation. The figure shows that our proposed method performs better than DARC in the target domain and other baseline methods.

Table 5: Comparison of DARAIL with DARC, 0.5 gravity.

|  | DARC Evaluation | DARC Training | Optimal in Target | DARAIL |
|---|---|---|---|---|
| HalfCheetah | $1686 \pm 392$ | $5721 \pm 463$ | $7559 \pm 782$ | $5485 \pm 592$ |
| Ant | $2058 \pm 553$ | $348 \pm 71$ | $3380 \pm 538$ | $990 \pm 12$ |
| Walker2d | $706 \pm 64$ | $936 \pm 158$ | $2830 \pm 482$ | $878 \pm 122$ |
| Reacher | $-13 \pm 1.3$ | $-11 \pm 1.9$ | $-7.2 \pm 0.3$ | $-12.2 \pm 0.5$ |

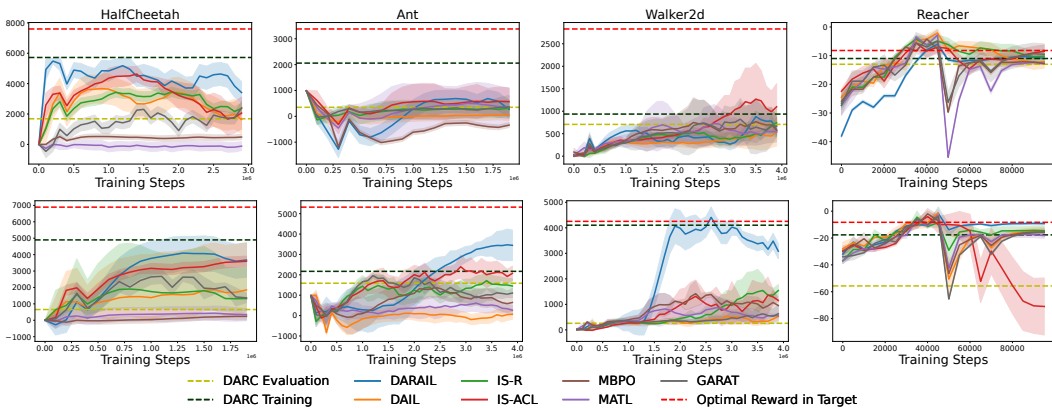

Figure 7: Training Curve of changing gravity setting. Top: target domain gravity×0.5, button: target domain gravity×1.5. Upper horizon line: DARC reward in the source domain. Lower horizon line: DARC reward in the target domain. The figures show the mean value of multiple runs and the standard deviation. The figure shows that our proposed method performs better than DARC in the target domain and other baseline methods.

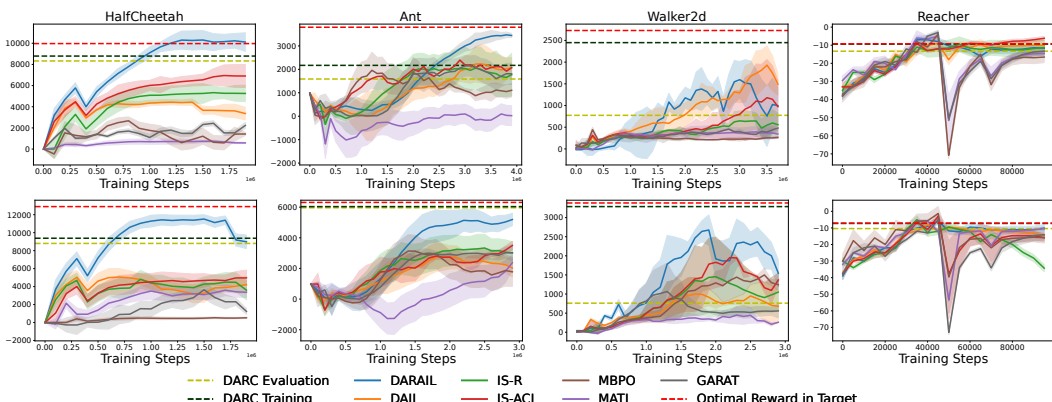

Figure 8: Training Curve of changing density setting. Top: target domain density×0.5, button: target domain density×1.5. Upper horizon line: DARC reward in the source domain. Lower horizon line: DARC reward in the target domain. The figures show the mean value of multiple runs and the standard deviation. The figure shows that our proposed method performs better than DARC in the target domain and other baseline methods.

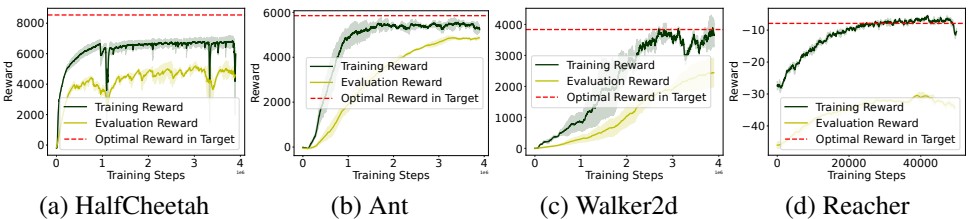

Figure 9: Training reward in the source domain, i.e. $\mathbb{E}_{\pi_{\text{DARC}}, p_{\text{src}}}[\sum_t r(s_t, a_t)]$, and evaluation reward in the target domain , i.e. $\mathbb{E}_{\pi_{\text{DARC}}, p_{\text{trg}}}[\sum_t r(s_t, a_t)]$, for DARC in four environments. Deploying trained DARC policy to the target domain will cause performance degradation.

Table 6: Comparison of DARAIL with baselines in off-dynamics RL, 0.5 gravity.

|  | DAIL | IS-R | IS-ACL | MBPO | MATL | GARAT | DARAIL |
|---|---|---|---|---|---|---|---|
| HalfCheetah | $3671 \pm 331$ | $3432 \pm 332$ | $4896 \pm 249$ | $12.2 \pm 42$ | $741 \pm 195$ | $3436 \pm 226$ | **4093** $\pm 1021$ |
| Ant | $970 \pm 16$ | $982 \pm 3.6$ | $984 \pm 77$ | $981 \pm 32$ | $980 \pm 46$ | $976 \pm 105$ | **990** $\pm 12$ |
| Walker2d | $541 \pm 315$ | $741 \pm 325$ | $1267 \pm 793$ | $724 \pm 423$ | $767 \pm 561$ | $823 \pm 458$ | **878** $\pm 122$ |
| Reacher | $-12.5 \pm 2.1$ | $-8.2 \pm 2.6$ | **-7.1** $\pm 2.6$ | $-16.2 \pm 0.1$ | $-13.6 \pm 0.1$ | $-13.7 \pm 3.5$ | $-12.2 \pm 0.5$ |

Table 7: Comparison of DARAIL with DARC, 0.5 density.

|  | DARC Evaluation | DARC Training | Optimal in Target | DARAIL |
|---|---|---|---|---|
| HalfCheetah | $8328 \pm 861$ | $8790 \pm 486$ | $9970 \pm 983$ | $10308 \pm 1042$ |
| Ant | $1587 \pm 224$ | $2170 \pm 195$ | $3798 \pm 341$ | $3472 \pm 245$ |
| Walker2d | $773 \pm 395$ | $2449 \pm 234$ | $2729 \pm 492$ | $1595 \pm 168$ |
| Reacher | $-13.3 \pm 1.2$ | $-9.4 \pm 1.5$ | $9.2 \pm 0.2$ | $-12.2 \pm 1$ |

Table 8: Comparison of DARAIL with baselines in off-dynamics RL, 0.5 density.

|  | DAIL | IS-R | IS-ACL | MBPO | MATL | GARAT | DARAIL |
|---|---|---|---|---|---|---|---|
| HalfCheetah | $4433 \pm 453$ | $5332 \pm 1063$ | $6951 \pm 1067$ | $740 \pm 172$ | $2676 \pm 315$ | $2437 \pm 213$ | **10308** $\pm 1042$ |
| Ant | $2233 \pm 809$ | $2050 \pm 892$ | $2396 \pm 96$ | $980 \pm 102$ | $1961 \pm 611$ | $2149 \pm 406$ | **3472** $\pm 245$ |
| Walker2d | **1930** $\pm 441$ | $646 \pm 226$ | $1180 \pm 789$ | $391 \pm 118$ | $441 \pm 59$ | $480 \pm 44$ | $1595 \pm 168$ |
| Reacher | $-12.2 \pm 1.8$ | $-13.3 \pm 4.2$ | $-13.2 \pm 1$ | **-11.7** $\pm 4.5$ | $-13.2 \pm 1.6$ | $-14.1 \pm 1.2$ | $-12.2 \pm 1$ |

Table 9: Comparison of DARAIL with DARC, 1.5 density.

|  | DARC Evaluation | DARC Training | Optimal | DARAIL |
|---|---|---|---|---|
| HalfCheetah | $8833 \pm 539$ | $9380 \pm 728$ | $6309$ | $11515 \pm 335$ |
| Ant | $5961 \pm 970$ | $6036 \pm 1345$ | $3288$ | $5193 \pm 463$ |
| Walker2d | $760 \pm 430$ | $3288 \pm 849$ | $3383$ | $2674 \pm 376$ |
| Reacher | $-10.4 \pm 0.4$ | $-7.3 \pm 1.3$ | $-7.1$ | $-10.2 \pm 2.1$ |

Table 10: Comparison of DARAIL with baselines in off-dynamics RL, 1.5 density.

|  | DAIL | IS-R | IS-ACL | MBPO | MATL | GARAT | DARAIL |
|---|---|---|---|---|---|---|---|
| HalfCheetah | $5057 \pm 766$ | $4814 \pm 524$ | $4966 \pm 727$ | $3598 \pm 706$ | $530 \pm 320$ | $3650 \pm 875$ | **11515** $\pm 335$ |
| Ant | $2738 \pm 781$ | $3335 \pm 1010$ | $3499 \pm 967$ | $2371 \pm 604$ | $3135 \pm 463$ | $3028 \pm 690$ | **5193** $\pm 463$ |
| Walker2d | $997 \pm 432$ | $1452 \pm 1036$ | $1950 \pm 198$ | $448 \pm 228$ | $1498 \pm 176$ | $1066 \pm 739$ | **2674** $\pm 376$ |
| Reacher | $-11.3 \pm 1.0$ | $-15.2 \pm 2.1$ | $-13.4 \pm 2.0$ | $-14.3 \pm 1$ | $-11.1 \pm 2$ | $-13.3 \pm 0.8$ | **-10.2** $\pm 2.1$ |

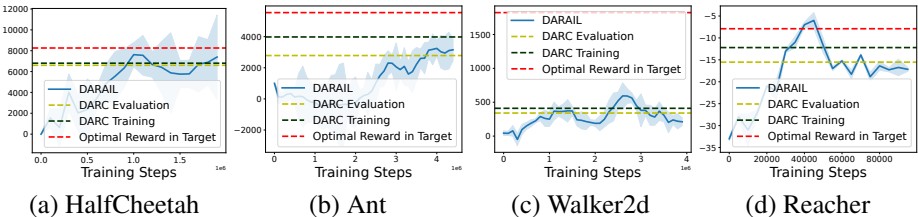

|                  |             |                  |               |
| (a) HalfCheetah  | (b) Ant     | (c) Walker2d     | (d) Reacher   |

Figure 10: Experiments of DARC and DARAIL on the intact source and broken target setting. We observe that the DARC does not have significant performance degradation. Also, we show that DARAIL can perform similarly to DARC in this setting.

## C.5 DARC training and evaluation performance on broken source setting

Figure 9 shows the performance of DARC trained in the source and evaluated in the target domain under broken source environment setting. The training reward is the reward obtained in the source domain, i.e. $\mathbb{E}_{\pi_{\text{DARC}},p_{\text{src}}}[\sum_t r(s_t,a_t)]$ and the evaluation is the reward deployed in the target domain, i.e. $\mathbb{E}_{\pi_{\text{DARC}},p_{\text{trg}}}[\sum_t r(s_t,a_t)]$. We observe the performance degradation in the figure 9. Empirically, we notice that the DARC policy performance in the source domain, $\mathbb{E}_{\pi_{\text{DARC}},p_{\text{src}}}[\sum_t r(s_t,a_t)]$, is close to the optimal reward in the target domain which matches with the DARC objective that DARC can generate target optimal trajectories in the source domain. However, deploying it to the target domain will result in performance degradation and a suboptimal reward due to the dynamics shift.

## C.6 Performance of DARAIL on broken target environment

We show the performance of DARAIL in the intact source and broken target environment setting in Figure 10 (the setting in DARC paper [3]). We observe that our method outperforms the DARC reward in the target domain, $\mathbb{E}_{\pi_{\text{DARC}},p_{\text{trg}}}[\sum_t r(s_t,a_t)]$. Also, we see that the performance of DARC in the source domain and target domain are very similar. Compared with the performance gap when the source environment is broken in Figure 9. As discussed, DARC works well when the assumption that the target optimal policy performs well in the source domain is satisfied. In the broken target setting, the target optimal policy can perform the same in the source domain.

Further, empirically, in the broken target setting, the DARC policy learns a near 0 value for the broken joint, which guarantees that the policy can generate similar trajectories in the two domains. Also, maximizing the adjusted cumulative reward in the source domain with a policy with a near 0 value for the broken joint is equivalent to maximizing the cumulative reward in the target domain. Thus, DARC perfectly suits the broken target setting. However, in the broken source setting and other more general dynamics shift cases, the target optimal policy might not perform well in the source domain. For example, in the broken source setting, the target optimal policy will perform poorly in the source domain as it loses one joint in the source domain. Another way to understand why DARC fails is that it learns an arbitrary value for the broken joint, which becomes detrimental in the target domain. However, this is just an artifact of the particular setting. As we discussed above, the intrinsic reason that DARC fails is the violation of the assumption.

## C.7 Performance of mimicking source optimal trajectories

In Figure 11, We compare our DARAIL, which uses DARC trajectories in the source domain as expert demonstrations and mimicking source optimal trajectories regardless of the target domain.

## C.8 Access to the target domain data compared to DARC.

Both DARC and DARAIL require some limited access to the target rollouts. In DARAIL, the imitation learning step only rolls out data from the target domain every 100 steps of the source domain rollouts, which is 1% of the source domain rollouts. We claim that more target domain rollouts will not improve DARC's performance due to its suboptimality, and DARAIL is better not because of having more target domain rollouts. We verify it by comparing DARC and DARAIL with the same amount of rollouts from the target domain in the broken source environment setting

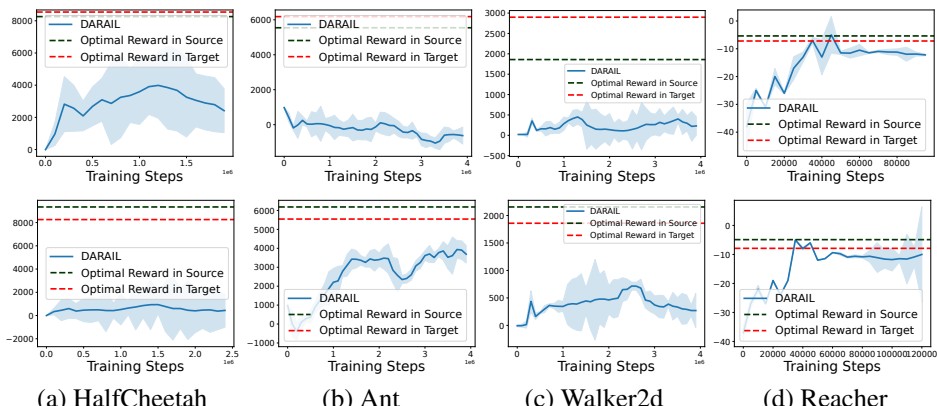

|   |   |   |   |
|---|---|---|---|
| (a) HalfCheetah | (b) Ant | (c) Walker2d | (d) Reacher |

Figure 11: Experiments on using source optimal policy as the expert demonstration instead of the DARC policy as the expert demonstration. Mimicking the source optimal trajectories will not receive a similar performance as mimicking DARC performance, and there is a big performance gap between the source optimal reward and imitation learning performance in the target domain.

in Tables 11 and 12. Specifically, we examine DARAIL with 5e4 target rollouts alongside DARC with 2e4 and 5e4 target rollouts. DARAIL has 5e3 target rollouts for the Reacher environment, while DARC has 3e3 and 5e3 rollouts. From the results, we see that increasing the target rollouts from 2e4 to 5e4 (or from 3e3 to 5e3 in the case of Reacher) does not yield a significant improvement in DARC's performance due to its inherent suboptimality. Notably, DARAIL consistently outperforms DARC when given comparable levels of target rollouts.

Table 11: Comparison with DARC with the same amount of rollout from the target. The number in the columns represents the amount of rollout from the target. More target domain rollout will not improve the DARC's performance further. Experiment on broken source setting.

|   | DARAIL 5e4 | DARC Evaluation 2e4 | DARC Training 2e4 | DARC Evaluation 5e4 | DARC Training 5e4 |
|---|---|---|---|---|---|
| HalfCheetah | $7067 \pm 176$ | $4133 \pm 828$ | $6995 \pm 30$ | $4037 \pm 798$ | $6988 \pm 27$ |
| Ant | $4752 \pm 872$ | $4280 \pm 33$ | $5197 \pm 155$ | $4342 \pm 42$ | $5207 \pm 172$ |
| Walker2d | $4366 \pm 434$ | $2669 \pm 788$ | $3896 \pm 523$ | $2538 \pm 802$ | $3782 \pm 510$ |

Table 12: Comparison with DARC with the same amount of rollout from target, on Reacher. The number in the columns represents the amount of rollout from the target. More target domain rollout will not improve the DARC's performance further. Experiment on broken source setting.

|   | DARAIL 5e3 | DARC Evaluation 3e3 | DARC Training 3e3 | DARC Evaluation 5e3 | DARC Training 5e3 |
|---|---|---|---|---|---|
| Reacher | $-13.7 \pm 0.9$ | $-26.3 \pm 3.3$ | $-11.2 \pm 2.9$ | $-29.7 \pm 4.1$ | $-10.2 \pm 1.2$ |

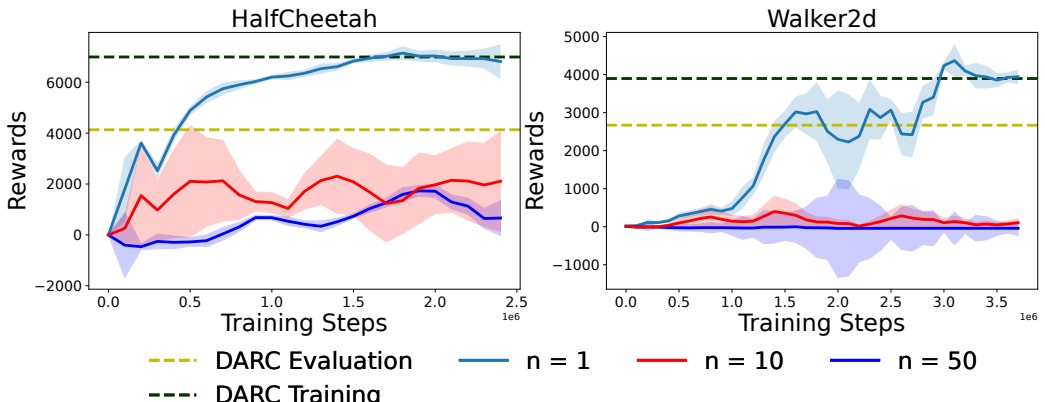

Figure 12: Experiment on how cumulative n-step importance weight performs on DARAIL. Per-step importance weight significantly outperforms using the last n-step multiplication of the importance weight.

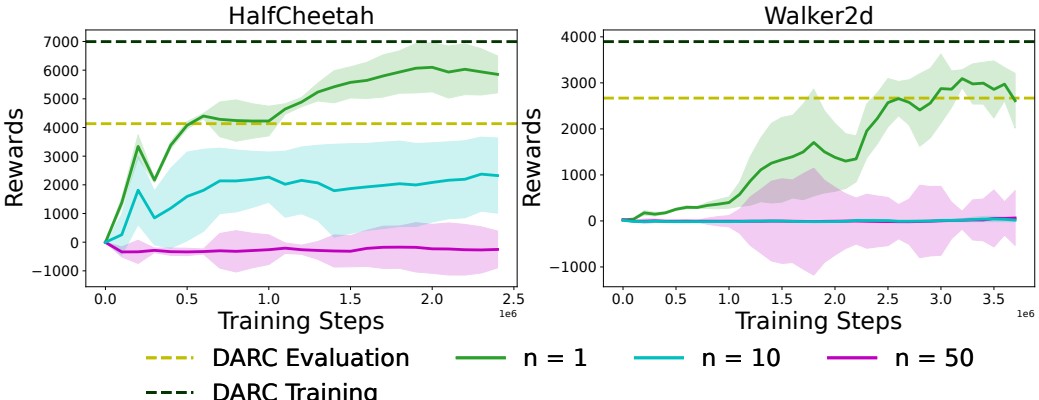

Figure 13: Experiment on how cumulative n-step importance weight performs on IS-R in broken source setting. Per-step importance weight significantly outperforms using the last n-step multiplication of the importance weight.

# D   Ablation Study

## D.1   Per-Step Importance Weight v.s Cumulative Importance weight

In our paper, to reduce the variance, we use the per-step importance weight $\frac{p_{\text{trg}}(s_t,s_{t+1})}{p_{\text{src}}(s_t,s_{t+1})}$ for the importance sampling method and DARAIL. Here, we compare the per-step importance weight with the cumulative n-step importance weight, which is the multiplication of the weight before time step $t$:

$$\rho_n(s_t, s_{t+1}) = \prod_{i=t-n}^{t} \frac{p_{\text{trg}}(s_{i+1}|s_i, a_i)}{p_{\text{src}}(s_{i+1}|s_i, a_i)}.$$

Note that here, the importance weight is the multiplication of the last n steps weight instead of the multiplication from $i = 0$ to $i = t$. Because the cumulative importance weight might have a *NaN* value due to the product. Thus, the optimization step for the imitation learning of DARAIL is as follows:

$$\max_{\zeta} \mathbb{E}_{p_{\text{src}},\zeta} \left[ \sum_t \rho_n(s_t, s_{t+1}) r(s_t, s_{t+1}) - (1 - \rho_n(s_t, s_{t+1})) \log D_\omega(s_t, s_{t+1}) \right].$$

Similarly, the objective of IS-R is:

$$\max_{\pi} \mathbb{E}_{p_{\text{src}}, \pi} \left[ \sum_t \rho_n(s_t, s_{t+1}) r(s_t, s_{t+1}) \right].$$

We compare the per-step importance weight and the cumulative n-step importance weight on DARAIL and IS-R. Specifically, we consider $n = [10, 50]$ for HalfCheetah and Walker2d, respectively. We show the results of DARAIL in Figure 12 and the results of IS-R in Figure 13. We see that the cumulative importance weight doesn't perform well on both methods and environments. In HalfCheetah, we can observe that the 10-step cumulative importance weight performs better than the 50-step one. And similar patterns appear in the Walker2d. Thus, we can conclude that per-step importance weight will have a lower variance and be more favorable in our experiment.

## D.2 Update Steps of Discriminator

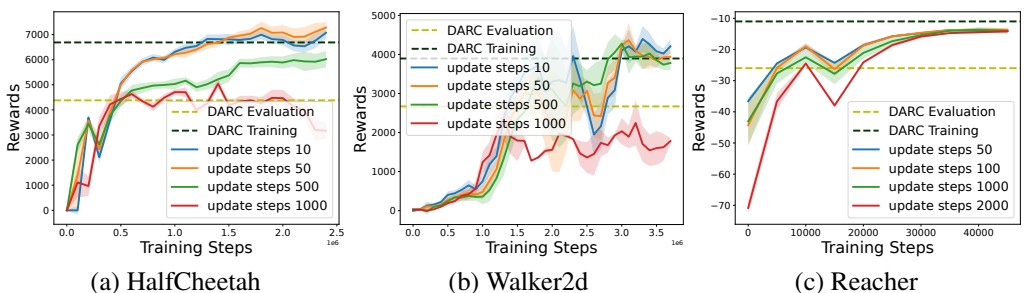

| (a) HalfCheetah | (b) Walker2d | (c) Reacher |

Figure 14: Experiment on the performance of DARAIL under different update steps of the discriminator in broken source setting.

In imitation learning, we alternatively update the generator and discriminator. In practice, we normally update the generator several steps and then update the discriminator once. The update steps, updating the discriminator every how many training steps, is a hyperparameter and is important in GAN training. The smaller the update steps are, the higher the update frequencies are. We tune the update steps and show the result of it in different environments. The best discriminator update step in HalfCheetah, Walker2d, and Reacher are 50, 50, and 1000, respectively. We varied the discriminator update steps in HalfCheetah and Walker2d in [10, 50, 500, 1000] steps, and the update steps in Reacher are [50, 100, 1000, 2000] steps. Figure 14 shows the effects of different discriminator update steps in the final performance. As we can see, for all three environments, a smaller update step (higher update frequency) is preferred as it can learn a better reward estimation. However, as we noticed, for example, for HalfCheetah and Walker2d, when the update step is 50, decreasing it to 10 will not further improve the performance.

## D.3 Increase the weight on the modified reward of DARC.

We tested DARC algorithm with modified reward $r(s_t, a_t) + \eta \Delta(s_t, a_t, s_{t+1})$ with $\eta > 1$ instead of $\eta = 1$. And the $\eta = 1$ is derived from the distribution matching objective in Eq.(3.3). We show the results in Figure 15 under the broken source environment setting. We can see that increasing $\eta$ will not increase the DARC performance in the target domain but will hurt the performance of DARC in the target domain.

## D.4 Hyperparameters

For a fair comparison, we tune the parameters of baselines and our method. The hidden layers of the policy and value network are [256,256] for the HalfCheetah, Ant, and Walker2d and [64,64] for Reacher. And the hidden layer of the two classifiers is [64] for the HalfCheetah, Ant, and Walker2d and [32] for Reacher. The batch size is set to be 256. We regularize the state by adding the running average of the state. We fairly tune the learning rate from $[3e-4, 1e-4, 5e-5, 1e-5]$. For those methods that require the importance weight $\rho$, we tune the update steps of the two classifiers trained to obtain the importance weight from $[10, 50, 100]$. We also add Gaussian noise $\epsilon \sim N(0, 1)$ to the

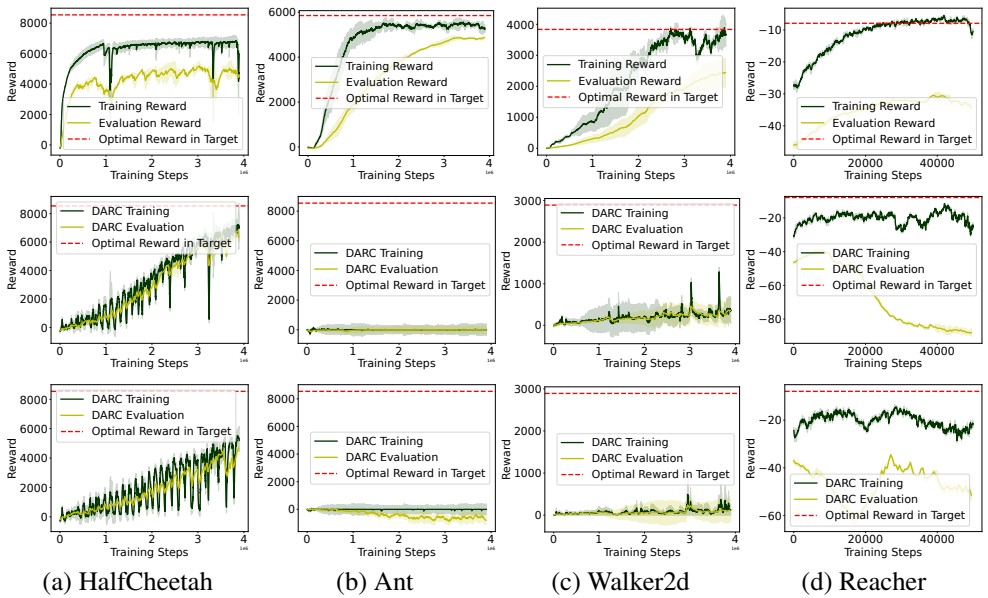

(a) HalfCheetah     (b) Ant     (c) Walker2d     (d) Reacher

Figure 15: Experiment of different $\eta$ in the modified reward $r(s_t, a_t) + \eta + \Delta(s_t, a_t, s_{t+1})$ for DARC in broken source environment setting. Top row: $\eta = 1$, middle row: $\eta = 1.5$ and button row: $\eta = 2$. We observe that increasing the $\eta$ will reduce the performance degradation in most cases, but it will also harm the performance of DARC in the target domain as increasing $\eta$ focuses more on making the DARC perform more similarly in both domains instead of maximizing the cumulative reward.

input of the classifiers for regularization, and the noise scale is selected from $[0.1, 0.2, 1.0]$. For the imitation learning component, the number of expert trajectories is 20. We further tune the update steps of the discriminator and add Gaussian noise to the input of the discriminator.

### D.5 Computation Resources

We run the experiment on a single GPU: NVIDIA RTX A5000-24564MiB with 8-CPUs: AMD Ryzen Threadripper 3960X 24-Core. Each experiment requires 12GB RAM and require 20GB available disk space for storage of the data.

## E Limitations

A potential limitation will be that we rely on DARC or similar methods to generate state pairs. An overly large dynamics shift, or data limitation may prevent us from obtaining high-quality state space data to imitate in the source domain. We do the experiment on the Mujoco environment instead of the real-world sim-2-real problem. We leave the investigation of this to future work.

