# OpenReview forum: "Off-Dynamics Reinforcement Learning via Domain Adaptation and Reward Augmented Imitation"
_NeurIPS.cc/2024/Conference — NeurIPS 2024 poster_

### Official Review · Reviewer_JwpF · 2024-06-24

**Soundness:** 3
**Presentation:** 2
**Contribution:** 3
**Rating:** 5
**Confidence:** 3

**Summary:**

This work introduces an improvement over the previous method (DARC) with theoretical and experimental contributions for solving the off-dynamics transfer learning problem.

**Strengths:**

- Using IL on states is a sensible idea to add
- The theoretical contribution is quite strong, as the authors were able to replace a very limiting assumption with a possibly more relaxed (and simpler to analyze) assumption
- There indeed appears to be a performance boost in the environments / transfer schemes tested.

**Weaknesses:**

- It's not clear how many samples are used as rollout in the target domain. If it is a non-trivial amount, then a baseline should be to run RL in the target domain with this sample budget.
- Similarly, DARC should also be allowed to have these additional samples in the target domain for proper comparison.
- Finally, the presentation quality poses a bit of a weakness. It is a bit difficult to read/analyze at times (see next section for suggestions to improve)
- Figure 3 does not appear to be a significant performance boost, can you comment on the purported gain?

**Questions:**

Questions about the paper:
In the appendix you show that lower discriminator update steps are better, stopping at 50. Can you explain why lower might be better; and why not decrease even further?
- You state a few times that you do not collect reward data from the target domain. Isn't this trivial since the reward function is identical to that of the source?


Various (some minor) presentation issues and references:
- suggest to abbreviate as "GAIL-O" or similar. Readers will likely know GAIL and thus understand the connection more quickly
- In main contributions, the first bullet point is quite unclear; can you re-word/shorten? Similarly, I believe you can remove all but the last sentence in bullet point 2.
- "works" -> "work" throughout
- Use of discounting: You use SAC with a discount factor, but $\gamma$ is missing from the main text and appendix in many equations. Can you include / comment on this? It appears the theory is only relevant for $\gamma=1$ which is okay but should be stated upfront.
- Relatedly, it may be beneficial to cite in Sect. 3.1 e.g. https://arxiv.org/abs/1805.00909 ;  https://arxiv.org/pdf/2005.01643
- Regarding the dynamics-shift problem, can you also comment on the relationship to e.g. https://proceedings.neurips.cc/paper_files/paper/2023/file/67dd6a41bf9539cffc0fc0165e4d0616-Paper-Conference.pdf ; https://ojs.aaai.org/index.php/AAAI/article/view/25817 ; https://proceedings.mlr.press/v155/pertsch21a/pertsch21a.pdf
- L80 "number of data" -> "amount of data"
- L84: "to mimic **an** expert"
- (nitpicking) L90: discussion of observations vs. states does not sound quite right; I believe the true distinction is in the POMDP framework
- Eq. 3.1 remove the extra parentheses around product.
- Eq. 3.3 Do you still need a min on the RHS? (I might be missing something, please explain if so!)
- the value of $c$ may be nice to see explicitly so that we can see independence on the dynamics and policy.
- L128: "Domian" -> "Domain"
- L135 "while we do not have much access to the target domain." can you be more specific?
- L150: "followed" -> "follows"
- L172: "Baye's" -> "Bayes'"
- Algorithm 1; maybe highlight which parts of the algorithm are distinct from DARC?
- Theorem 4.1: I believe it would be helpful to either (a) give a more formal version of the theorem with all definitions (define H, D_omega, is $\gamma$ missing?) or (b) make it an "informal" theorem, with the full result in the appendix.
- Which version of the environments are you using?
- L304: "DARAIL's ??? on Different..." is there a missing word (Effect)?
- Figures: Can you please include a shaded region for the DARC eval/training reward lines?
- Figure 3: DARRIL- -> DARAIL?



----
Overall, I believe the paper needs a significant clean-up of writing and additional experiments and discussion to merit a higher score.

**Limitations:**

As stated above, two important limitations the authors can further comment on are (a) use of samples in the target domain and (b) gain in performance.

---

> ### Author Rebuttal · Authors · 2024-08-07
>
> Thank you for your valuable time and effort in providing detailed feedback on our work. We hope our response will fully address all of your points.
>
> ---
> >How many samples are used as rollouts in the target domain?
>
> Our imitation learning step rolls out data from the target domain every 100 steps of the source domain rollouts, which is 1\% of the source domain rollouts.
>
> ---
> >Additional samples from the target domain for DARC.
>
> DARAIL does have more target domain rollouts. However, as mentioned in the general response, more rollouts of the target domain data is not the reason why DARAIL works. More rollouts of the target domain will not improve DARC performance, according to our analysis of why DARC fails in more general settings in the general response. As mentioned in the third bullet point in the general response, we compare DARC and DARAIL with the same amount of rollouts from the target domain in Tables 4 and 5 in the attached PDF, indicating that more target domain rollouts does not yield significant improvement in DARC's performance due to its inherent suboptimality. Notably, DARAIL consistently outperforms DARC even when subjected to comparable levels of target rollouts.
>
> ---
> >Explanation of Figure 3.
>
> Figure 3 in the paper is an ablation study showing that our method works well under different scales of off-dynamics shift. Specifically, we conducted experiments on Ant-v2, and the target environment is broken with probability 0.2,0.5,0.8, respectively. The smaller the broken probability is, the smaller the dynamic shift is.
>
> ---
> >Smaller discriminator update steps are better.
>
> In GAIL or, more generally, GAN, the discriminator serves as the local reward function for the policy training. The smaller discriminator update steps (higher update frequency) probably lead to better discriminator training (better local reward estimation). In our experiments, we did the grid search for the parameters and noticed that further decreasing the update steps (increasing the update frequency) would not improve the performance; it is probably already well-trained, so we stopped that at 50. We will provide a more thorough grid search results to justify this in the future.
>
> ---
> >Presentation comments.
>
> We thank the reviewer for the detailed comments that help us improve the readability of the paper. We will follow your suggestions in our revision. For several questions, we briefly answer as follows:
> 1) **Missing discount factor $\gamma$ in equations.** We will state clearly in the revision that we use $\gamma = 1$ in the analysis of the error bound.
> 2) **Discuss of the related work.** [1] is most related to our problem. However, they assume that the optimal policy in different domains has similar stationary state distribution, thus, they learn a policy in the source domain and regularize the state distribution in two domains. This method is similar to DARC, which tries to learn a policy that behaves similarly in both domains. However, when the assumption is violated, the learned policy will have performance degradation when deployed to the target domain, similar to DARC. In contrast, our method doesn’t require such an assumption and we propose one more imitation learning step to transfer the policy to avoid performance degradation. [2] and [3] are designed to solve a slightly different problem that transfers prior knowledge to solve a new task instead of just considering the dynamic shift between the tasks.  We will add the discussion of these works to the related work in the revision.
> 3) **Discussion of observations vs. states in L90.** The observation here represents the state trajectories we obtain from the environment, which is a general terminology from imitation learning from observation. We use imitation learning from observation to distinguish from generative adversarial imitation learning (GAIL) which uses $(s_t,a_t)$ as the expert demonstration. Meanwhile, imitation learning from observation purely learns from the observation (state observation) from the environment, i.e. $(s_t,s_{t+1})$.
> 4) **Right-hand side needs a min for Eq 3.3.** The right-hand side of Eq 3.3 is the mathematical expansion of the reverse KL divergence loss. We will add a min to that to make it clearer.
> 5) **Meaning of do not have much access to the target domain.** The limited access means that we cannot freely roll out infinite data from the target domain, and we can only roll out a small amount of data from the target domain. This is a standard setup shared by many off-dynamics problems, including DARC. Also, we assume we do not query the target domain reward during the training. As discussed in the first bullet point, we roll out the states and actions from the target domain data every 100 steps of the source domain rollouts. We will make it more clear in the revision.
>
> We appreciate the reviewer's feedback. We hope we have thoroughly addressed your concerns. We are willing to answer any additional questions.
>
> ---
> **References**
>
> [1] State Regularized Policy Optimization on Data with Dynamics Shift.
>
> [2] Accelerating Reinforcement Learning with Learned Skill Priors.
>
> [3] Utilizing Prior Solutions for Reward Shaping and Composition in Entropy-Regularized Reinforcement Learning.

---

> > ### Comment · Reviewer_JwpF · 2024-08-12
> >
> > Thank you to the authors for responding to my questions. I believe you have somewhat addressed my concerns and hopefully the camera-ready version can include some of this discussion. As stated before, the paper will benefit significantly from a careful rewriting, which I hope the authors will do. I've accordingly raised my score to 5.

---

> > > ### Author Response · Authors · 2024-08-12
> > >
> > > Thank you very much for your positive feedback! We will revise our paper according to your constructive reviews.
> > >
> > > Best,
> > >
> > > Authors

---

### Official Review · Reviewer_VFUe · 2024-07-09

**Soundness:** 2
**Presentation:** 2
**Contribution:** 2
**Rating:** 5
**Confidence:** 4

**Summary:**

This paper provides solution to a specific kind of domain adaptation, where the source action space is a subset of the target action space. In this setup, the source policy might fail in the target domain, because the policy would output arbitrary values for the action dimensions only active in the target domain, but not in source. To resolve this, the paper disentangles the effect of actions, and considers the source domain's policy's state trajectories as the optimal trajectories to imitate in the target domain. By applying imitation learning from observation, they can learn a target policy to reproduce the state transitions from the source policy, but using the action space from the target domain. One trivial solution to this is that the target policy would learn the source policy's actions, just with a 0 output (or a value that leads to no effect) for the extra dimensions in the target domain. The paper outperforms the prior work DARC on its specialized environment setup, but unlikely to be a general method for other off-dynamics setups.

**Strengths:**

- The paper provides a good overview of the prior work DARC, and helped to understand their method without having to go through DARC.
- The experimental evaluation consists of several baselines, and DARAIL generally outperforms them, achieving similar to source performance. This shows that the imitation learning from observation pipeline is working.
- They provide code for reproducibility.

**Weaknesses:**

## Evaluation on a very specific kind of dynamics change
The major issue with this work is its limited evaluation where the only kind of off-dynamics shift is freezing the 0-index value to 0 in the action in the source domain. Ideally, if the action support of source is a subset of the target domain, then the same policy should be perfectly fine in the target domain. Yet, this paper shows that the same policy underperforms in the target domain. The only reason why this might be happening is because the DARC policy learns an arbitrary value for the 0-th dimension of action torque, which is inconsequential in the source domain, however, becomes detrimental in the target domain. The paper never discusses this obvious reason for this failure, and the only thing a method needs to do is to learn to always output 0 for the 0th-index, for it to work in the target domain.

DARAIL is specifically constructed so that it learns to output 0 and works well for this artificial environment setup, because it **assumes the source trajectory to be the optimal trajectory in the target domain**. But this is a major assumption not made in the original DARC paper. In fact, I would argue (and I am open to change my mind on this if proven otherwise) that DARAIL does not work in the off-dynamics settings considered in DARC paper. For instance, consider ant which is crippled in the target domain but not the source domain (from DARC paper). Considering the source ant's trajectories as optimal could be detrimental for the target crippled ant, because most of the state transitions $(s_t, s_{t+1})$ in source domain are not even possible in the target domain. The crippled ant requires a different optimal policy to be learned, which cannot be obtained by just imitating the source state trajectories, which DARAIL would do.

In order to show that DARAIL is actually a general solution to off-dynamics RL, the paper should include experiments on:
- All the same setups provided in DARC paper.
- At least one more example of off-dynamics RL other than just freezing the 0-index of action.

## Paper Presentation
The paper is often written in a confusing manner and there are several quality issues in presentation. The following need to be corrected in writing:
- Figure 6 (b) for Ant and 6 (d) for Walker are the exact same training curves. Please fix this.
- The paper never discusses why DARC "fails" is because of the 0-th action dimension being an arbitrary output in the target domain, because it was never trained to be 0 during training.
- L2-3: "performance degradation especially when the source domain is less comprehensive than the target domain": "less comprehensive" is vague.
- In L23-25, the paper mentions that "in domains such as medical treatment [1] and autonomous driving [2], we cannot interact with the environment freely as the errors are too costly". However, in Algorithm 1 L7, it assumes access to the target environment for rolling out.
- L41-42: "the source domain has less action support, which is normally a harder off-dynamics RL problem." Why is this a harder problem? If anything, target domain having a less action support would mean that the agent needs to adapt to more adverse off-dynamics conditions.
- L64-65: "DARAIL relaxes the assumption made in previous works that the optimal policy will receive a similar reward in both domains." — What does this mean, because L78 says that both the source and target domains have the same reward function? In fact, DARAIL considers the source trajectories as optimal trajectories for the target domain.
- What does it mean for the two domains to have the same reward function, when the dynamics are different? Does the paper assume the reward function is of the form R(s) only and not R(s, a, s')? Any assumptions made in the problem setup should be clearly written.
- In Eq 3.1 and 3.2, the product term $\Pi_t$ should include the last terms inside the bracket.
- L110-111: This cannot be right. The optimal policy should be proportional to the exponential of the cumulative returns, not reward at time step t.
- Figure 1 (a): It is unclear how this figure suggests that there is any **performance degradation**. Since the dynamics are different, it is possible that the optimal reward in the target domain could be just different from the optimal reward in the source domain — comparing these two does not make sense. It only works in this paper because of the artificial formulation that the source action space is a subset of the target action space. The correct comparison would be evaluating $pi_{DARC}$ in the target domain against some $\pi_{optimal}$ for the target domain, both in the target domain. The training reward obtained in the source domain does not tell us anything for general off-dynamics setups.
- L151-166: This paragraph is hard to follow.
- Figure 7: Some baselines are not trained for the same number of steps.

## Method is overly complicated for the problem setup
The paper makes a questionable assumption that when $\pi_{DARC}$ is optimal for source domain, the trajectories it obtains are also optimal in the target domain and thus, we should achieve similar state trajectories in the target domain. While this is a very restricted assumption, let's say for the sake of the argument, this assumption makes sense for certain useful setups. Even then, why learn $\pi_{DARC}$ using any influence from the target trajectories at all? Why not just learn $\pi_{src}$ from the src reward only, while disregarding the target trajectories and not applying DARC at all. And then we can transfer that policy using Gaifo to the target environment. What is the idea of doing DARC at all, when anyway the source policy is to going to be used for imitation learning from observations in the target domain? If DARC on src is really important, it should be experimentally validated with convincing arguments.

**Questions:**

I have listed suggestions to improve in the weakness section above. There are a lot of changes in the paper presentation that are necessary, but my major concern would still be about the limited applicability of the proposed problem setup and the method not even best suited for that proposed problem.

**Limitations:**

The paper mentions a couple of limitations in the appendix section, not in the main paper. The paper does not discuss several limitations of their problem setup's applicability, their method's potential fallacies on the original DARC environments, and their limiting assumption of source trajectories being optimal for the target domain.

---

> ### Author Rebuttal · Authors · 2024-08-07
>
> Thanks for your valuable time and detailed feedback. We hope our response will fully address all of your points.
>
> ---
> >Evaluation on a very specific kind of dynamics change.
>
> 1) In the general response, we provide additional experiment results under DARC setting and more general off-dynamics settings to show that our method works well beyond the specific dynamics change. In the second bullet, we clarify why DARC fails in the broken source and intact target settings theoretically and empirically.
>
> 2)  **DARAIL is specifically constructed so that it learns to output 0.** This is not true. As mentioned in the general response, Table 1 in the provided PDF shows DARAIL works for the general off-dynamics setting other than broken action. Indeed, if we know the difference in dynamics, we can “simply” learn/set 0 for 0th index. However, we do not assume we know beforehand. We will add this discussion to our revision.
>
> 3) **DARAIL does not work in the off-dynamics settings considered in DARC.**  It’s not true that most of the state transitions  $(s_t,s_{t+1})$ generated by DARC in the source domain are not possible in the target domain in DARC settings. DARC learns a near 0 value for the 0th index, as mentioned in the second bullet in the general response. Table 3 in the provided PDF shows that DARAIL performs better than the DARC evaluation in the target domain in DARC setting.
>
> ---
> >Method is overly complicated for the problem setup.
>
> First, **we do not assume that “when $\pi_{DARC}$ is optimal for the source domain, the trajectories it obtains are optimal in the target domain.**” Instead, we notice that  $\pi_{DARC}$ generates the trajectories in the source domain that **resemble** the target optimal ones, as this is the DARC’s learning objective. So we transfer $\pi_{DARC}$ to the target domain instead of mimicking the source optimal policy. We provide additional experiments that mimic the source optimal policy instead of DARC trajectories on two settings in Tables 2 and 3 in the PDF. DARAIL significantly outperforms that, probably because the optimal source domain trajectories might not perform well in the target domain.
>
> Second, we want to clarify further that we do not assume that the source trajectories generated by DARC are **exactly** the optimal trajectories in the target domain. Instead, we utilize DARC’s learning objective to learn a policy that generates trajectories in the source domain that **resemble** the target optimal ones. Moreover, we considered DARC's training error in the first term of the error bound in our analysis.
>
> ---
> >Paper presentation questions
> 1) **Performance degradation happens when the source domain is less comprehensive.** We want to clarify that less comprehensive means that the support in the source is less than the target, such as our setting. However, we agree that this is not the only case when performance degradation happens, as suggested in Table 1. We will clarify this point in our revision.
>
> 2) **Why is it a harder problem when the source domain has less support?** In the domain adaptation problem, if the source domain/training data has larger support or covers more data distribution than the target, it would be easier to generalize to the target domain because the training data has seen all possible target actions/data. In off-dynamics RL settings, if the target domain has less support, we can restrict the action support in the source domain. For example, in the DARC setting, whose target environment has less support, the DARC policy learns to output near 0 value for the 0th-index, and the problem becomes optimizing the policy in the source domain such that the DARC policy outputs 0 for the 0th-index. From this perspective, DARC is restricting the action support in the source domain to align the target domain action support. However, on the opposite, where the source has less action support, such methods don’t work well, and we need to account for the unseen/unsupported action in the target domain.
>
> 3)  **DARAIL relaxes the assumption made in previous works that the optimal policy will receive a similar reward in both domains.** As we discussed in the general response, the assumption made by DARC guarantees that DARC’s performance and generated trajectories in the target are similar to those in the source domain. This assumption may not hold in other off-dynamics settings. We want to relax this assumption by not restricting the performance of target optimal policy in the source domain. Note that the DARC source domain trajectories resemble the optimal target ones, according to DARC’s learning objective. Thus, we use imitation learning to transfer such policy. And the experiment shows the effectiveness of our method when their assumption fails.
>
> 5) **Two domains have the same reward function.** It means that $r_{\text{src}}(s_t,a_t) = r_{\text{trg}}(s_t,a_t)$ not $r(s)$. We will make it clearer in the revision.
>
> 6) **Clarifications on Figure 1(a) and performance degradation.** To show the performance degradation empirically in more general settings, we employ DARC in the target domain and compare it with its performance in the source in Figure 1(a). Here, the motivation is, according to the DARC’s objective, that the DARC policy on source should achieve a similar performance (and a similar trajectory distribution) to the optimal policy on target. But we agree, empirically, it may not be true. So, we followed the reviewer’s suggestion to add the reward for the optimal policy on target to show the performance degradation better in Figure 1, Table 2 and 3. In contrast, little performance degradation happens in the DARC setting. We will update the figures and tables accordingly in our revision.
>
> 7) **Not trained for the same steps.** We train the baselines until convergence. Though we believe the result would not change much with more training steps, we will train the baselines to the same number of steps for fair comparison in the revision.

---

> > ### Comment · Reviewer_VFUe · 2024-08-11
> >
> > Thank you for adding the new experiments on off-dynamics settings and DARC settings. The results of the intact source and broken target environment (DARC) are convincing to me, and they make sense to me now. The DARC policy learned in the source is on the modified reward and thus will tend to imitate the target data, thus generating trajectories that resemble optimal state sequences in the target domain. So, applying DARAIL should not deteriorate your starting point of DARC. This also explains why using just an optimal source policy is not enough, and you need to start with a DARC policy. I am raising my score to 5 and would be willing to increase further if the following questions are answered.
> >
> > 1. I appreciate you confirming that DARC fails in your original setting because it learns an arbitrary value of zero-index. Can you provide similar qualitative reasoning for why DARC fails in the gravity-change setting but DARAIL works perfectly fine? I understand that the underlying reason is that the assumption of the DARC paper might not hold, but can you discuss how exactly that turns out in the half-cheetah/reacher environments? And why does DARAIL (which also uses a DARC-like policy) not suffer from the same issues and perform so well?
> >
> > 2. In the gravity experiment, DARC significantly underperforms for HalfCheetah (1544 v/s 5818) but works almost as well for Reacher.
> > - What is the reasoning here, and how many seeds are these results?
> > - The poor performance of DARC in HalfCheetah is quite extreme; the gap is even larger than the experiments in the original paper. This is impressive but also surprising. I don't understand why DARC would fail so significantly here.
> > - Actually, now that I think about it, does Reacher even get affected by the coefficient of gravity?
> >
> > 3. In the experiment on optimal source policy imitation, did you exactly run Algorithm 1, with the only change being that you replace Step 2: [Call DARC], you do not use the modified reward $r_{\text{modified}} = r(s_t, a_t) + \Delta r(s_t, a_t, s_{t+1})$, but use the environment reward only: $r_{\text{unmodified}} = r(s_t, a_t)$ ? If not, what is the "Mimic Source Optimal Policy" experiment?
> >
> > 4. If I understand correctly, DARAIL exploits some rollouts in the target environment to match the DARC's source trajectory distribution. This results in a DARAIL policy that is more suitable to be deployed in the target environment than DARC's learned policy. But, could this effect of making the effect of target "stronger" be done in another way, like learning a modified DARC policy with $r_{\text{modified}} = r(s_t, a_t) + \alpha \Delta r(s_t, a_t, s_{t+1})$, where $\alpha > 1$? I know the authors added a rebuttal experiment with more target rollouts to DARC. But that only makes the $\Delta r$ term more accurate. It does not increase its influence in the DARC's learned policy. If my understanding is correct here, then this simple solution should be a baseline in this paper. If the authors can try it out, that would be great. The implementation change is as easy as adding a coefficient and experimenting with values like 1.5, 2, 10, etc. — depending on the reward scale. But if not, I would be happy with an explanation of why this would not work. Why is DARAIL, which is much more complex and comes with additional requirements for online environment rollouts with DARC policy, necessary?
> >
> > 5. There are several points in the paper presentation that were not addressed (understandably due to the lack of space in rebuttal response). Some of these are important. Now that there is no word limit, can you make sure everything is replied to?

---

> > > ### Author Response · Authors · 2024-08-12
> > >
> > > 1, **Qualitative reasoning for why DARC fails in the gravity-change setting, but DARAIL works perfectly fine?** As mentioned in the previous general response, DARC works when the performance of the target optimal policy is similar in the source domain. This is hardly true when we have a different gravity in general. For example, in simple cases like Pendulum and InvertedPendulum, gravity parameters directly change the angle of the pole, the velocity of the cart, etc. Thus, it’s possible that the target optimal policy can lead to a different angle and velocity, which could be more likely to fall for the pole in the source domain. Intuitively, in this gravity-changing setting, applying the DARC policy to the target domain might generate very different trajectories in the target domain than the source ones, causing performance degradation. For example, in the InvertedPendulum, we run experiments modifying the gravity coefficient from 1.0 to 10. DARC receives optimal rewards in the source domain, but only 40\% of the source domain rewards in the target domain. In this setting, the reward is defined as whether the pole stays on the cart at each step and the trajectory ends once the pole falls. This means that the length of DARC target trajectories is only 40\% of DARC source trajectories.
> > >
> > > **DARAIL works in this setting.** DARAIL works well in this setting because we propose using imitation learning to transfer these trajectories to the target domain instead of directly deploying DARC to the target domain. By doing so, we do not require the assumption that the dynamic of the target optimal policy performs similarly in the source domain. Though DARAIL utilizes DARC trajectories in imitation learning, it doesn’t mimic the DARC policy directly but mimics the DARC trajectories in the state space in the source domain, which avoids deploying DARC directly to the target domain.
> > >
> > > 2, **DARC significantly underperforms for HalfCheetah but works almost well for Reacher.** First, the experiment results are averaged over 5 runs. Second, in Reacher, changing the gravity may not create a significant dynamic shift compared to the HalfCheetah, though the coefficient of gravity is modified from 1.0 to 0.5. Besides gravity, we also noticed that modifying the “friction” of the Reacher configuration file from 1.0 to 0.5 or 2.0 will not cause very large performance degradation. How they affect the performance degradation is determined by how the parameters like gravity and friction affect the dynamics model. For tasks that depend more on gravity, like HalfCheetah, the performance degradation is more obvious. We will have more experiments in different environments, dynamic shifts, and scales of the shifts, including gravity, friction, and density, in our revision.
> > >
> > > 3, **Mimicking the source optimal policy.** Our experiment is exactly what you describe that using the unmodified reward $𝑟_{unmodified}=𝑟(𝑠_𝑡,𝑎_𝑡) $ in Step 2: [call DARC].
> > >
> > > 4, **Modifying DARC $\Delta r$’s scale.**  Thanks for the suggestion. As mentioned, DARC training constrains the DARC policy to behave similarly in the two domains, and the  $\Delta r$ in the modified reward can be regarded as the constraint or the regularization term during the training. Thus, increasing the $\alpha$ will give a stronger constraint for policy optimization, making DARC behave more similarly in both domains. However, that doesn’t guarantee that DARC policy will perform well in the target domain. Moreover, increasing the $\alpha$ (stronger regularization) might affect the training to focus more on the constraint instead of maximizing the reward. We further run experiments on HalfCheetah with the broken source, the intact target environment, and the gravity-changing setting (from 1.0 to 0.5) to justify this. The following table shows the results of DARC in HalfCheetah with different $\alpha$. As we increase $\alpha$ from 1.0 to 2.0, we observe that the DARC reward in the source and target are getting closer for both cases. However, the training has a strong oscillation due to the regularization. Due to the regularization, though DARC evaluation can benefit from increasing $\alpha$, the DARC training performance will be affected to be close to the DARC evaluation performance. In this case, DARAIL still outperforms DARC with $\alpha = 2$. We will experiment more and add this baseline on the different $\alpha$ in our revision.
> > >
> > > |            |DARC Training, alpha = 2 | DARC Evaluation, alpha = 2| DARC Training, alpha = 1 | DARC Evaluation, alpha = 1|  DARAIL|
> > > |----|-----------|---------------------------------------------|-----------------|--|-----------------|
> > > |Changing Gravity                | 2438   | 2307 | 5828   | 1544  | 5818|
> > > |Broken Environment           | 6171  | 5955  | 6995 |  4133  | 7067|

---

> > > ### Author Response · Authors · 2024-08-12
> > >
> > > 5, **Paper presentation.**
> > >
> > > **Typo.** Thanks for the suggestion on the paper writing. We will fix the typo in writing, including the Figure, $\prod_t$ in Eq 3.1/3.2. Also, the optimal policy is proportional to the exponential of the cumulative rewards, not reward at time step t in Lines 110-111.
> > >
> > > **More explanation to Line 151-166 regarding the reward estimator $R_{AE}$.** The imitation learning step iteratively trains a discriminator (maximizing Eq 3.5) and then updates policy with the signal provided by the discriminator, which is $-log_{D_{\omega}}(s_t,s_{t+1})$. We can view this as a ‘local reward estimator,’ and the policy is updated with $-log_{D_{\omega}}(s_t,s_{t+1})$ in imitation learning instead of the reward $r(s_t,a_t)$. However, this estimation can be biased and inaccurate in generative adversarial training, and the problem is even more severe due to the off-dynamics shift. Similarly, in the off-policy evaluation (OPE) of offline bandit and RL, this bias and inaccurate reward estimation exist. And doubly robust estimator is employed in this case to correct the reward estimation with importance weight term. Motivated by the doubly robust estimator, we propose a surrogate reward function that uses the source domain reward $r_{src}(s_t,a_t)$ (as the reward function is the same across the domains) and the importance weight between the dynamics to correct the local reward estimator $-log_{D_{\omega}}(s_t,s_{t+1})$, which is similar to the doubly robust estimator that uses $\rho (r-\hat{r})$ to correct the reward estimator $\hat{r}$. Here, the $\rho$ is the importance weight.
> > >
> > > We believe we have covered all the questions regarding paper presentations. If there is still more clarification needed, please let us know. Thank you again for the suggestions.

---

### Official Review · Reviewer_LdFt · 2024-07-12

**Soundness:** 3
**Presentation:** 2
**Contribution:** 3
**Rating:** 7
**Confidence:** 3

**Summary:**

The paper considers the off-dynamic RL setting and identifies that existing approach, DARC, fails to obtain the true optimal policy in the target MDP. Leveraging ideas from imitation learning literature, the paper proposes a learning objective that takes into the account the dynamic shift between the source and target MDPs. The objective is theoretically sound and the conducted experiments show that the proposed method, DARAIL, is able to account for severe dynamic shift in MuJoCo environments.

**Strengths:**

- The experiments show that DARAIL is able to account for severe dynamic shift and is not too sensitive with clipped importance sampling ratio.
- The relationship between the proposed estimator and doubly robust estimator is interesting.

**Weaknesses:**

- The notation can be defined more clearly.
	- On page 3, line 106: Is $\tau$ a trajectory sequence of state-action pairs, or just sequence of states?
		- What generates this?
	- On page 3, Eq (3.2): Is $\tau_{\pi^*}^{trg}$ Is this trajectory generated by $\pi^*$ in the target domain?
- It will be preferable if the paper can provide explanations on why DARC fails to address for the dynamic shift.

**Questions:**

**Questions**
- On page 4, lines 120-122: Is this always true, or just may? If I understand correctly since the dynamics are "unknown" it may recover the wrong target policy (i.e. $\pi^* \neq \pi^{trg}$)?
- DARC does not need any online learning while DARAIL does correct? I believe this should be highlighted if that is the case.
- On page 4, lines 151-152: This shared reward is an assumption correct? What if this is not true either?

**Possible typos**
- Page 2, line 128: Domain

**Limitations:**

**Limitations**
- The experiments are purely in MuJoCo so I am curious what can happen in other domain.

---

> ### Author Rebuttal · Authors · 2024-08-07
>
> We first thank the reviewer for their time and comments. We now address your questions point by point.
>
> 1, **Notation of $\tau$.** $\tau$ represents the trajectory sequence of state-action pairs, so it is a notation for the trajectories. For example, we use $\tau_{\pi_{\theta}}^{\text{src}}$  to represent the trajectories generated by the policy $\pi_{\theta}$ in the source domain.
>
> 2, **Why DARC fails to address dynamics shift.** First, as mentioned in the general comment, DARC works well under some dynamic shifts when the assumption about the dynamic shift made in the DARC paper is satisfied, i.e. the optimal policy for the target domain is also a good policy in the source domain. DARC works well in this setting because the objective of DARC can also be viewed as maximizing the cumulative reward in the source domain under a constraint that the learned policy performs similarly in two domains. However, in more general off-dynamics settings. This assumption is not satisfied. DARC will be suboptimal in the target domain as DARC does not perform similarly in the two domains, leading to performance degradation in the target domain.
>
> Our method is motivated by DARC, which generates trajectories in the source domain that resemble the optimal ones in the target domain. We propose to use imitation learning to transfer DARC trajectories to the target domain so that deploying the policy to the target domain will not receive such performance degradation.
>
> 3, **On page 4, lines 120-122, recover the wrong target policy.**
> We would appreciate any clarifications on the definition of $\pi_{trg}$ in the second part of the question. To answer the first part, the matching of the distribution is the main goal of the DARC’s learning objective. Under the unknown dynamics and unknown dynamic shift, DARC tries to learn a policy that generates trajectory distribution similar to the target trajectories from the target optimal policy. But the policy $\pi_{DARC}$ will not be the same as the target optimal policy due to the dynamics shift. Here, we only use the $\pi^*$, which is the target optimal policy, in our analysis of DARC’s failure.
>
> 4, **DARC does not need any online learning, while DARAIL does.** DARC does need online learning and they sample $(s_t,s_{t+1})$ from the target domain. Similarly, our method requires sampling from the target domain. In both cases, the amount of data rollouts from the target is significantly smaller than the rollouts from the source. The details can be referred to the third bullet point of the general response.
>
> 5, **Assumption about the shared reward**. In the off-dynamic RL setting, we specifically focus on the setting where the two domains only differ in the dynamics but share reward functions and state/action space. If this assumption is not true, this becomes a different domain adaptation problem, and we might need other techniques to adapt to the shift in the reward function.
>
> 6, **Experiment on other settings instead of Mujoco.** We follow the recent work in off-dynamics RL and experiment on the MuJoCo environment, which is easy to access and analyze. We will also add new experiments on other environment settings in our revision.
>
>
> We appreciate the reviewer's feedback. We hope we have thoroughly addressed your concerns. We are willing to answer any additional questions.

---

> > ### Comment · Reviewer_LdFt · 2024-08-12
> >
> > Thank you for the response, regarding (3) you have addressed my question. I will keep the score as is.

---

> > > ### Author Response · Authors · 2024-08-12
> > >
> > > Thank you very much for your reply!
> > >
> > > Best,
> > >
> > > Authors

---

### Official Review · Reviewer_mGy8 · 2024-07-30

**Soundness:** 3
**Presentation:** 4
**Contribution:** 3
**Rating:** 6
**Confidence:** 3

**Summary:**

The authors introduced the approach Domain Adaptation and Reward Augmented Imitation Learning (DARAIL) for off-dynamics RL. The work aim at generating the same trajectories in the target domain as expert trajectories learned via DARC in the source domain. With GAIL-style framework and reward augmentation, DARAIL reached SOTA in bench mark off-dynamiacs environments. DRAIL does not assume the learned expert policy in the source domain to be close to the optimal policy in the target domain, and the authors are able to give a theoretical error bound of the method according to the dynamics shift scale.

**Strengths:**

1. The problem of off-dynamics RL is very meaningful, given some target environments and data are not easily accessible.
2. The shortcoming of DARC is clearly stated. Movitation is clear.
3. The theoretical bound proof is neat.
4. Sufficient experiments on both on baseline comparison, sensitivity analysis and dynamics shift sclaes.

**Weaknesses:**

1. Typo in Line #81, expectation should be over p_trg
2. Typo in Appendix C.2 Line #537 (C.1)
3. Typo in Appendix C.2 Line #540, #546
4. There is no comparison experiements of directly using (s,s’) from the expert trejectories in the source domain for DARAIL.
5. Freezing the 0-index action is not a good way to validate the effectiveness of the method on dynamics shift. Experiemnts on ,ore diverse dynamics shift environments are needed.

**Questions:**

How would DARC trajectories quality influence the overall imitation learning performance in the target domain?

**Limitations:**

Did not test on more complicated dynamics shifts.

---

> ### Author Rebuttal · Authors · 2024-08-07
>
> We first thank the reviewer for their time and comments. We now address your concerns point by point.
>
> ---
> > How would DARC trajectory quality influence the overall imitation learning performance in the target domain?
>
> We use trajectories generated by the DARC in the source domain as the expert trajectories for imitation learning, which resemble the optimal trajectories in the target domain, according to the DARC’s objective. In general, the imitation learning performance should be similar to the DARC performance in the source domain, with the latter one expected to perform similarly with the target optimal policy if the DARC learning error is small. The better the DARC trajectory quality, the better performance the resulting policy from imitation learning can achieve. Overall, our analysis in section 4 shows that a good DARC trajectory quality will have a smaller learning error in the first term of the error bound.  Further, we want to emphasize that our method is not particularly limited by the performance of DARC because our method can be viewed as a general framework that uses imitation learning to transfer trajectories from the source domain to the target domain as long as we can obtain trajectories that resemble the optimal trajectories on the target.
>
> ---
> > There are no comparison experiments of directly using $(s,s’)$ from the expert trajectories in the source domain for DARAIL.
>
> We assume that by “$(s, s’)$ from the expert trajectory in source,” the reviewer means using the optimal trajectory in the source. We provide such experiment results on HalfCheetah and Reacher in Table 2 and 3 in the attached PDF. We first train a policy in the source domain (do not consider the dynamics shift) and use imitation learning to transfer it to the target domain. The results show that DARAIL outperforms the method that uses optimal trajectories in the source domain as the expert policy for imitation learning because DARC can generate trajectories that resemble the optimal ones in the target domain.
>
>
> ---
> > Experiments on other off-dynamics shifts.
>
> We provide additional experiment results on HalfCheetah and Reacher in Table 1 in the provided PDF. The off-dynamics shift is constructed by modifying MuJoCo's configuration files. We modify the coefficient of gravity from 1.0 to 0.5 for the target domain. Our results show the effectiveness of our method. In our revision, we will add more experiments on these more general cases to demonstrate the performance of DARAIL.
>
> We thank the reviewer for their detailed suggestions for improving the readability of our paper. We will follow them closely, fix all the typos, and clarify the writing of the paper.

---

### Author Rebuttal · Authors · 2024-08-07

We want to thank all reviewers for their time and constructive feedback on our paper. Since some reviewers referred to similar concerns, we would like to make a general response to address these questions.

---
> Experimental results on more general off-dynamics settings and more baselines (R_mGy8 and R_VFUe).

In the attached file, we provided additional experiment results on HalfCheetah and Reacher to validate our algorithm in general off-dynamics settings. The off-dynamics shift is constructed by modifying MuJoCo's configuration files. We modify the coefficient of gravity from 1.0 to 0.5 for the target domain. Our results in Table 1 of the attached PDF file show the effectiveness of our method. In our revision, we will add more experiments on these more general cases to demonstrate the performance of DARAIL.

We also validate the DARAIL in intact source and broken target environments (DARC settings). The results are shown in Table 3 of the attached file. We can see that DARAIL inherits the good performance of DARC in this setting. Considering the limited settings in DARC and the more general settings in our experiment, **DARAIL will not degrade the performance of DARC in the source intact and target broken settings and will improve the performance in more general settings.**

Further, as suggested by R_mGy8 and R_VFUe, in Table 2 and 3 of the attached file, we compare with the performance of the target optimal policy to better show the suboptimality of DARC. Also, we compare DARAIL with mimicking the $(s_t,s_{t+1})$ of the optimal trajectories in the source domain and show the superiority of DARAIL.

---
> Regarding “why DARC fails” in more general off-dynamics cases (R_LdFt and R_VFUe).

We will explain it from theoretical and empirical aspects and compare the two settings: broken source and intact target v.s. intact source and broken target.

Theoretically, as stated in Lemma B.1 in the DARC paper, the objective of DARC is equivalent to training a policy (maximizing the cumulative reward) in the source domain under a constraint that DARC behaves similarly in both domains and thus receives similar rewards. Also, in the analysis of the error bound in DARC, they made a strong assumption in Assumption 1 that **optimal policy for the target domain should also receive a similar reward in the source domain**, that is $E_{p_{\text{src}}, \pi^*}[\sum_t r(s_t,a_t)] -E_{p_{\text{trg}}, \pi^*}[\sum_t r(s_t,a_t)] \leq 4R_{max} \sqrt(\epsilon/2)$, where the $\pi^*$ is the optimal policy in the target domain, the $R_{max}$ is the maximum reward of a trajectory, $\epsilon$ is the slack term and the error bound depends on the $\epsilon$. In the setting where the source is intact and the target is broken (same as in DARC paper), the optimal policy for the target domain gives a 0 value for the 0th-index. Thus, deploying this policy in the intact source domain will receive the same reward. And the assumption here is perfectly satisfied with $\epsilon=0$. Thus, the error bound for the optimal DARC policy is 0, as analyzed in Theorem 4.1 in the DARC paper.

Empirically, we notice that under the DARC setting, the DARC policy learns a near 0 value for the 0th-index. This guarantees that the policy can generate similar trajectories in the two domains. Also, maximizing the adjusted cumulative reward in the source domain with a policy with a near 0 value for the 0th-index is equivalent to maximizing the cumulative reward in the target domain. Thus, DARC perfectly suits the source intact and target broken setting.

However, in the source broken and target intact setting, and also other more general off-dynamics settings, the optimal policy in the target environment might not perform well in the source domain. It may even have a large performance degradation, like in the source broken and target intact settings, the policy loses one action dimension in the source domain. Thus, the assumption made in the DARC paper might not hold or, formally speaking, the $\epsilon$ there can be very large, leading to performance degradation of DARC in the target domain.

In particular, intuitively, in the source broken and target intact settings, we agree with R_VFUe that the DARC fails as it learns an arbitrary value for the 0-th dimension of action torque, which becomes detrimental in the target domain. However, this is just an artifact of the particular setting. As we discussed above, the intrinsic reason that DARC fails is the violation of the assumption. In Table 1 of the provided PDF, we demonstrate DARC’s failure in other off-dynamics settings where the dynamics shift is not induced by the broken action.

---
> Regarding access to target domain data (R_JwpF and R_VFUe).

Both DARC and DARAIL require some limited access to the target rollouts. In DARAIL, the imitation learning step only rolls out data from the target domain every 100 steps of the source domain rollouts, which is 1\% of the source domain rollouts. We claim that more target domain rollouts will not improve DARC’s performance due to its suboptimality, and DARAIL is better not because of having more target domain rollouts. We verify it by comparing DARC and DARAIL with the same amount of rollouts from the target domain in Tables 4 and 5 in the attached file. Specifically, we examine DARAIL with 5e4 target rollouts alongside DARC with 2e4 and 5e4 target rollouts. DARAIL has 5e3 target rollouts for the Reacher environment, while DARC has 3e3 and 5e3 rollouts. From the results, we see that increasing the target rollouts from 2e4 to 5e4 (or from 3e3 to 5e3 in the case of Reacher) does not yield a significant improvement in DARC's performance due to its inherent suboptimality, as mentioned in the second bullet. Notably, DARAIL consistently outperforms DARC when given comparable levels of target rollouts.

---

### Decision · Program_Chairs · 2024-09-25

**Decision:**

Accept (poster)

**Comment:**

This paper proposes an RL method for off-dynamics RL (adapting to new environments) by combining reward augmentation (as done in prior work) with imitation (novel contribution). The reviewers generally lean towards accepting the paper: scores = 7/6/5/5. The reviewers appreciated the importance of the problem, the clear motivation, thorough experiments, and the theoretical results. Several reviewers had writing suggestions, which the authors have promised to make. During the rebuttal period, the authors and reviewers had a thorough discussion, which seems to have resolved most of the concerns the reviewers initially had about the paper (e.g., with additional experiments). Taking this together, I recommend that the paper be accepted.